# Towards a Foundation Model Approach for Causal Graph Learning

## Abstract

Due to its human-interpretability and invariance properties, Directed Acyclic Graph (DAG) has been a foundational tool across various areas of AI research. However, DAG learning remains highly challenging, due to its super-exponential growth in computational cost and identifiability issues, particularly in small-sample regimes. To address these two challenges, we leverage the recent success of transformers and develop a foundation model approach for discovering multiple DAGs across tasks. In particular, we propose Attention-DAG (ADAG), a novel attention-mechanism-based architecture for learning multiple linear Structural Equation Models (SEMs). ADAG learns the mapping from observed data to both graph structure and parameters via a nonlinear attention-based kernel, enabling efficient multi-task generalization of the underlying linear SEMs. By formulating the learning process across multiple domains as a continuous optimization problem, the pre-trained ADAG model captures the common structural properties as a shared low-dimensional prior, thereby reducing the ill-posedness of downstream DAG tasks in small-sample regimes. We evaluate our proposed approach on benchmark synthetic datasets and find that ADAG achieves substantial improvements in both DAG learning accuracy and zero-shot inference efficiency. To the best of our knowledge, this is the first practical approach for pre-training a foundation model for unsupervised DAG learning, representing a step toward more efficient and generalizable down-stream applications in causal discovery.

## 1 Introduction

Causality plays a fundamental role in explaining the underlying mechanisms of systems in many scientific decision-making domains (Pearl et al., 2000; Sachs et al., 2005; Lu et al., 2021; Subbaswamy & Saria, 2020). This has led to significant interest within the machine learning community in developing advanced methods for causal discovery. A common approach to model causal relationships is to identify causal models among a set of random variables in the form of Directed Acyclic Graphs (DAGs), which offer a compact, interpretable, and theoretically grounded representation of the underlying data-generating process. However, learning DAGs from observational data remains highly challenging due to the super-exponential space of possible graph structures, inherent identifiability issues, and data scarcity in real-world applications. Moreover, most existing approaches operate on a per-task basis, lacking the ability to generalize across tasks or domains. As a result, there is growing interest in developing foundation models that can transfer knowledge across causal tasks.

Inspired by the recent success of foundation models and their capacity to encode vast amounts of transferable information, we propose to address the high computational cost and poor performance in low-sample regimes by pre-training a DAG-learning foundation model that generalizes across tasks to infer DAGs accurately and efficiently. Specifically, we introduce a novel attention-mechanism-based formulation for learning Structural Equation Models (SEMs). As illustrated in Figure 1, the key of our approach is to define a nonlinear kernel mapping using the attention blocks, which takes observational data as the input and the corresponding weighted adjacency matrices as the output. During the pre-training phase, the model is trained across multiple tasks, which each task treated as a DAG discovery problem (recovering a hidden DAG from a set of data observations). As such, the weighted adjacency matrix is inferred in an unsupervised way, and the model is capable to capture both the structural causal relationships and the associated causal mechanisms. The multi-task training and attention mechanism-based blocks were known to possess advantages in inferring

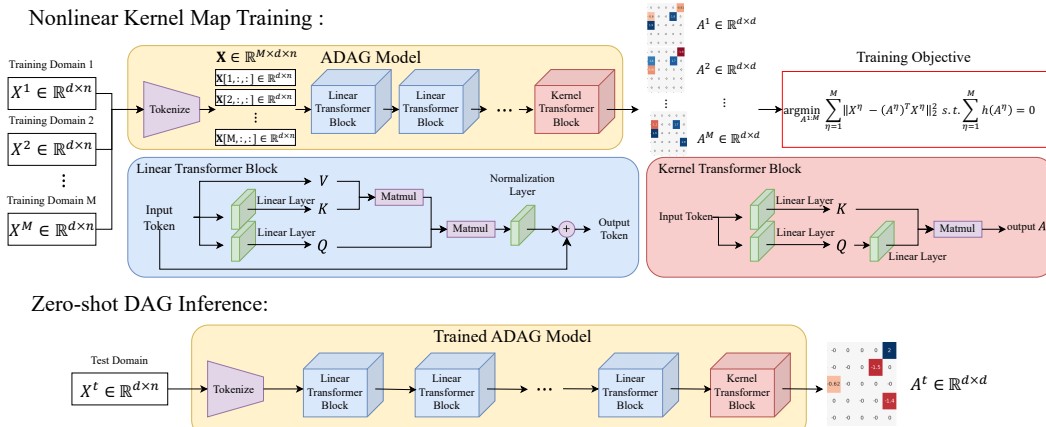

Figure 1: The schematic of the proposed training and inference procedures for unsupervised DAG learning. Given data from multiple domains, we begin by training a nonlinear kernel mapping from data to the corresponding hidden DAG. The trained model is then capable of zero-shot DAG inference, allowing it to predict weighted adjacency matrices when given data from new test domains.

shared low-dimensional structure across tasks and applying such a structure in the downstream test tasks (Lu & Yu, 2025). Therefore, our proposed ADAG architecture is anticipated to reduce the ill-posedness of individual DAG estimation problems and improve recovery of both graph structure and parameters, especially in data-scarce settings. Furthermore, once trained, our model enables zero-shot inference on previously unseen tasks, offering significant improvements in both accuracy and efficiency for downstream causal discovery. Notably, our framework does not require ground truth graphs, as the DAG kernel mapping is learned automatically using a data reconstruction loss, which differs from supervised DAG learning approaches (Li et al., 2020) such as the amortized methods (Lorch et al., 2022; Ke et al., 2022).

To the best of our knowledge, this is the first work to propose a practical approach for pre-training a foundation model for unsupervised DAG learning. Existing efforts to integrate causality with foundation models have very different focuses, such as causal inference (Zhang et al., 2024a) or semantic information extraction from existing large language models (Ban et al., 2023; Wan et al., 2024; Wu et al., 2024), as they do not directly train foundation models to estimate DAG structures and underlying causal model parameters from observed data alone. While several prior works (Chen et al., 2021; Lu & Gao, 2023; Zhang et al., 2017; Zhou et al., 2022) have developed advanced algorithms for multi-task DAG learning, they do not generalize to unseen tasks without further optimization or training steps. In contrast, our approach does not require ground-truth graphs and supports zero-shot generalization to new tasks. This sets the stage for a new direction in causal discovery: building scalable, pre-trained foundation models capable of generalizing structural knowledge across domains.

**Major Contributions.** 1. We propose a novel attention-mechanism-based formulation for DAG learning, that learns a nonlinear kernel mapping from observational data to the underlying causal graph structure and associated parameters. 2. We obtain a foundation model for DAG learning. This model captures shared low-dimensional structures and enables zero-shot inference on unseen tasks. 3. We demonstrate that our method significantly outperforms both state-of-the-art DAG learning baselines in inference efficiency and accuracy, especially in small-sample regimes.

## 2 RELATED WORKS

**Attention Mechanism for Inverse Problems.** In recent years, the transformer based on the attention mechanism has been increasingly adopted to tackle diverse scientific problems. It has been found that the attention mechanism is capable of modeling complex dependencies within sequential or structured data, leading to novel applications in various domains (Guo et al., 2023; Ovadia et al., 2024; Yu et al., 2024; Evangelista et al., 2023; Chen et al., 2023; Cao, 2021). Unlike many investigations on applying the attention mechanism for forward problems (Vladymyrov et al., 2024; Lu et al., 2024; Zhang et al., 2024b), causal discovery and graph structure learning in general fall in the regime of inverse problems. In inverse problems, the objective is not merely to predict future outputs, but to infer the underlying relationship that generates the observed data. To the authors' best

knowledge, the attention mechanism and transformer in general have been relatively underexplored in the context of inverse problems, despite their potential to complement existing deep learning strategies (Afkham et al., 2021; Evangelista et al., 2025).

**Optimization-Based DAG Learning.** Existing DAG learning approaches either quantify conditional independence relationships among variables through statistical tests or search for the optimal DAG by maximizing a predefined score using various search strategies. A notable shift was introduced by Zheng et al. (2018), who proposed reformulating the DAG learning problem from a combinatorial optimization task into a constrained continuous optimization problem, allowing for the use of gradient-based optimization methods. Subsequent works have improved various aspects of the continuous optimization framework (Yu et al., 2019; Bello et al., 2022; Ng et al., 2020; Yu et al., 2021; Khemakhem et al., 2021; Yin et al., 2024; Zheng et al., 2020; Lachapelle et al., 2019; Deng et al., 2024). Despite these advances, DAG learning remains NP-hard (Chickering, 2002), with the number of possible DAGs growing super-exponentially with the number of variables. Although the continuous optimization framework improves tractability, it does not eliminate the high computational cost. Moreover, existing approaches require a sufficient amount of data that accurately captures all underlying causal dependencies and their performance degrades when data is scarce in real-world scenarios. These two challenges motivate us to develop pre-trained models that are expressive enough to encode rich and transferable representations from available training data and generalize to unseen data, enabling efficient and accurate DAG inference even in small-data regimes.

**Beyond Single Domain DAG Learning.** Since our training procedure involves recovering the underlying mechanisms between data observations and DAGs by jointly performing DAG learning across multiple training domains, the problem naturally falls within the multi-task learning setting. We therefore review existing works on multi-task DAG learning. In particular, Chen et al. (2021) assumes that data from different tasks are generated by distinct DAGs that share a common topological order. Lu & Gao (2023) and Zhang et al. (2017) assume that the underlying DAG structure is shared across tasks, while the data generation mechanisms (i.e., causal mechanisms) vary.

**Amortized DAG learning.** Amortized DAG learning seeks to train models that can directly predict a DAG from data. Prior work includes supervised approaches where models are trained to output binary DAG structures (Lorch et al., 2022; Ke et al., 2022). Montagna et al. (2024) focus on bivariate causal models, providing analyses of identifiability and generalization, while Scetbon et al. (2024) propose a two-step framework that first infers causal orderings with an amortized model and then estimates the causal graph conditioned on those orderings. A common characteristic of existing approaches is their reliance on supervised training to produce binary DAGs directly.

## 3 MATHEMATICAL FORMULATION

### 3.1 LINEAR STRUCTURAL EQUATION MODEL FOR MULTI-DOMAIN DATA

We begin by introducing our proposed attention-mechanism-based formulation for DAG learning. Given a set of $d$ random variables $X = [X_1, X_2, \cdots, X_d] \in \mathbb{R}^d$, the linear Structural Equation Model (SEM) with additive noise is defined as:

$$X = A^T X + E \tag{1}$$

where $A \in \mathbb{R}^{d \times d}$ is the weighted adjacency matrix representing the DAG. The entries of $A$ encode both the causal structure and the causal mechanisms, such that a nonzero entry $A[i, j] \neq 0$ indicates a causal link $X_i \rightarrow X_j$. The noise vector $E = [E_1, E_2, \cdots, E_d] \in \mathbb{R}^d$ consists of mutually independent exogenous noise variables.

In our setting, we assume the availability of $M$ domains of observations over the same set of $d$ variables, denoted as $\mathcal{D} = \{X_{1:d}^\eta\}_{\eta=1}^M$. The corresponding SEM for the $\eta$-th domain is

$$X_{1:d}^\eta = (A^\eta)^T X_{1:d}^\eta + E \tag{2}$$

where $A^\eta \in \mathbb{R}^{d \times d}$ denotes the adjacency matrix of the DAG in the $\eta^{\text{th}}$ domain. for each domain $\eta$, $n$ observations of $X_{1:d}^\eta$ are collected. We denote the collected data in domain $\eta$ as $\{X_{1:d}^\eta(j)\}_{j=1}^n$. Our goal is to infer the corresponding $A^\eta$ from the data $\{X_{1:d}^\eta(j)\}_{j=1}^n$ on the $\eta-$th domain.

### 3.2 NONLINEAR MAPPING FROM DATA TO GRAPH STRUCTURE AND PARAMETERS

To recover the domain-specific adjacency matrix $A^\eta$ from the observed data $X_{1:d}^\eta$, we propose leveraging the expressive power of attention mechanisms to learn the underlying mapping between the

observed data and the corresponding causal structure. To this end, we first model the weighted adjacency matrix $A^\eta$ as a function of the data $X_{1:d}^\eta$. Given the collected data in domain $\eta$, $\{X_{1:d}^\eta(j)\}_{j=1}^n$, we first transfer it to tokens $\mathbf{X}^\eta(1:n)$:

$$\mathbf{X}^\eta(1:n) = \Big(\mathbf{X}^\eta(1); \mathbf{X}^\eta(2); \cdots ; \mathbf{X}^\eta(n)\Big) = \Big(X_{1:d}^\eta(1); X_{1:d}^\eta(2); \cdots ; X_{1:d}^\eta(n)\Big) \in \mathbb{R}^{d\times n}. \quad (3)$$

Here, each token represents the data from a variable, and it consists a vector of size $n$, concatenating the information from all $n$ samples on this domain. The weighted adjacency matrix $A^\eta$ can then be modeled as a function of the tokens $\mathbf{X}^\eta(1:n)$, dependent on trainable parameters $\Theta$:

$$A^\eta = A[\mathbf{X}^\eta(1:n);\Theta]. \quad (4)$$

We point out that although a linear SEM is considered in this work, the kernel map from the input $\mathbf{X}^\eta(1:n) \in \mathbb{R}^{d\times n}$ to the output, the weighted adjacency matrix $A^\eta \in \mathbb{R}^{d\times d}$, is highly nonlinear. To capture this complex nonlinear relation, we parameterize the function $A^\eta[\cdot;\Theta]$ by designing an $L$-layer attention model:

$$\begin{aligned} \mathbf{H}_{\text{in}}^\eta =&\mathbf{H}_{(0)}^\eta := \mathbf{X}^\eta(1:n) \in \mathbb{R}^{d\times n}, \\ \mathbf{H}_{(l)}^\eta :=&\text{Attn}[\mathbf{H}_{(l-1)}^\eta; \theta_l]\mathbf{H}_{(l-1)}^\eta + \mathbf{H}_{(l-1)}^\eta \in \mathbb{R}^{d\times n}, 1 \le l \le L, \\ A^\eta :=&\text{Attn}[\mathbf{H}_{(L)}^\eta; \theta_{\text{out}}] \in \mathbb{R}^{d\times d}, \end{aligned} \quad (5)$$

where the attention block writes:

$$\text{Attn}[\mathbf{H}_{(l-1)}^\eta; \theta_l] = \sigma\Big(\frac{1}{\sqrt{d}}\mathbf{H}_{(l-1)}^\eta \mathbf{W}_l^Q (\mathbf{W}_l^K)^T (\mathbf{H}_{(l-1)}^\eta)^T\Big) \in \mathbb{R}^{d\times d}. \quad (6)$$

In the $l^{\text{th}}$ attention block, the trainable parameters are $\theta_l = \{\mathbf{W}_l^Q \in \mathbb{R}^{n\times k}, \mathbf{W}_l^K \in \mathbb{R}^{n\times k}\}$, and $\sigma(\cdot)$ is the activation function[1]. In the last layer, we output the weighted adjacency matrix as:

$$A^\eta = \text{Attn}[\mathbf{H}_{(L)}^\eta; \theta_{\text{out}}] = \mathbf{W}_{\text{out}}^{P,x}\sigma\Big(\frac{1}{\sqrt{d}}\mathbf{H}_{(L)}^\eta \mathbf{W}_{\text{out}}^Q (\mathbf{W}_{\text{out}}^K)^T (\mathbf{H}_{(L)}^\eta)^T\Big), \quad (7)$$

where the trainable parameters are $\theta_{\text{out}} = \{\mathbf{W}_{\text{out}}^{P,x} \in \mathbb{R}^{d\times d}, \mathbf{W}_{\text{out}}^Q \in \mathbb{R}^{n\times k}, \mathbf{W}_{\text{out}}^K \in \mathbb{R}^{n\times k}\}$. By substituting the above formulation into the SEM in equation 2, we have:

$$X_{1:d}^\eta(1:n) = A^T[X_{1:d}^\eta(1:n);\Theta]X_{1:d}^\eta(1:n) + E(1:n), \quad (8)$$

with $\Theta = \{\theta_l\}_{l=1}^L \cup \theta_{\text{out}}$.

## 3.3 ATTENTION MECHANISM-BASED DAG LEARNING

Similar to prior continuous optimization-based DAG learning methods, we propose to learn a nonlinear kernel map by solving the following optimization problem:

$$\min_{\Theta := \{\theta_{\text{out}}, \theta_{1:L}\}} \sum_{\eta=1}^M \|X_{1:d}^\eta(1:n) - A^T[X_{1:d}^\eta(1:n);\Theta]X_{1:d}^\eta(1:n)\|_F^2 \quad (9)$$

$$\text{s.t. } h(A[X_{1:d}^\eta(1:n);\Theta]) = 0, \forall \eta \in \{1, 2, \cdots, M\}.$$

Here, $h(A[X_{1:d}^\eta(1:n);\Theta]) = h(A^\eta) = \text{tr}(e^{A^\eta \circ A^\eta}) - d = 0$ is the acyclicity constraint proposed in Zheng et al. (2018), which ensures that $A^\eta$ represents the weighted adjacency matrix of a DAG. Our method does not impose restrictions on the choice of acyclicity constraint; alternative formulations of the DAG constraint from Bello et al. (2022) and Zhang et al. (2022) can also be used.

As shown in equation 9, our goal is to learn a nonlinear kernel map from data observations to weighted adjacency matrices by jointly performing DAG learning across multiple data domains. This makes the optimization problem substantially more challenging than in the single-task DAG learning setting. Although recent methods for single DAG learning (Bello et al., 2022; Ng et al., 2020; Yu et al., 2021) have improved efficiency by avoiding time-consuming iterative optimization, they are either not directly applicable to our pre-training scenario or yield suboptimal performance.

---

[1] $\sigma(\cdot)$ is set to be the identity activation function in the paper, because it enables a more efficient implementation using linear attention (Liu & Yu, 2025). However, other activation functions can also be used.

To ensure the accuracy of the weighted adjacency matrices used in learning the nonlinear kernel map, we adopt the augmented Lagrangian method during training and solve a sequence of optimization subproblems. With a slight abuse of notation, we re-write the optimization problem as:

$$\max_{\alpha \in \mathbb{R}} \min_{\Theta} \mathcal{L}_{\text{rec}}\big(X_{1:d}^{1:M}(1:n); A^{1:M}\big) + \frac{\rho}{2} \sum_{\eta=1}^{M} |h(A^\eta)|^2 + \alpha \sum_{\eta=1}^{M} h(A^\eta),$$

$$\mathcal{L}_{\text{rec}}\big(X_{1:d}^{1:M}(1:n); A^{1:M}\big) := \frac{1}{2n} \sum_{\eta=1}^{M} \sum_{j=1}^{n} \sum_{i=1}^{d} \Big(X_i^\eta(j) - A^T\big[X_{1:d}^\eta(1:n); \Theta\big][i,:]X_{1:d}^\eta(j)\Big)^2. \quad (10)$$

where we denote $A[X_{1:d}^\eta(1:n); \Theta]$ as $A^\eta$ for simplicity in equation 10. For each time, we solve the following optimization with the updated $\alpha$ value:

$$\Theta_\alpha^* = \arg\min_{\Theta} \mathcal{L}_{\text{rec}}\big(X_{1:d}^{1:M}(1:n); A^{1:M}\big) + \frac{\rho}{2} \sum_{\eta=1}^{M} |h(A^\eta)|^2 + \alpha \sum_{\eta=1}^{M} h(A^\eta), \quad (11)$$

then update $A^{1:M}$ and $\alpha$ with equation 12.

$$(A^\eta)_\alpha^* \leftarrow A[X_{1:d}^\eta(1:n); \Theta_\alpha^*], \qquad \alpha \leftarrow \alpha + \rho \sum_{\eta=1}^{M} h\big((A^\eta)_\alpha^*\big). \quad (12)$$

We summarize the proposed algorithm in Algorithm 1.

---

**Algorithm 1** Attention-DAG (ADAG) Training Process

1: **Input:** Training domain data $X_{1:d}^{1:M}(1:n)$, initial guesses of $\Theta_0$ and $\alpha_0$, progress rate $c \in (0,1)$, tolerance $\epsilon > 0$, threshold $\omega > 0$
2: **for** $t \leftarrow 1$ **to** $n$ **do**
3:     Solve $\Theta_{t+1} \leftarrow \arg\min_{\Theta} \mathcal{L}_{\text{rec}}\big(X_{1:d}^{1:M}(1:n); A^{1:M}\big) + \frac{\rho}{2} \sum_{\eta=1}^{M} |h(A^\eta)|^2 + \alpha_t \sum_{\eta=1}^{M} h(A^\eta)$,
    with $\rho$ such that $\sum_{\eta=1}^{M} h(A_{t+1}^\eta) < c \sum_{\eta=1}^{M} h(A_t^\eta)$.        ▷ Use Adam optimizer
4:     Update $A_{t+1}^{1:M} \leftarrow A[X_{1:d}^{1:M}(1:n); \Theta_{t+1}]$.
5:     Update $\alpha_{t+1} \leftarrow \alpha_t + \rho \sum_{\eta=1}^{M} h(A_{t+1}^\eta)$.
6:     **if** $\sum_{\eta=1}^{M} h(A_{t+1}^\eta) < \epsilon$ **then**
7:         $\hat{\Theta} = \Theta_{t+1}$ and break.
8:     **end if**
9: **end for**
10: **return** the optimal parameters $\hat{\Theta}$ for the nonlinear kernel map $A[\cdot; \Theta]$.

---

**Nonlinear SEM Extension.** Although we formulate our attention-mechanism-based DAG learning problem under the linear SEM assumption, the idea can be easily extended to the nonlinear SEM setting by first applying nonlinear transformations to the input $X_{1:d}^\eta(1:n)$ before multiplying it with weighted adjacency matrices. In fact, we can use shared attention layers for both the nonlinear kernel map and the nonlinear transformation, and multiply $A$ with $\mathbf{H}_{(L)}^\eta$ instead of $X_{1:d}^\eta(1:n)$ in equation 8.

**Discover the Prior to Enhance Identifiability.** While most single-task DAG learning methods mainly consider sufficient rank problems ($n \gg d$), in this work we consider a more challenging scenario with small observed data in each domain. In the latter case, the inverse problem may become under-determined, making the learning non-identifiable. As shown in Lu & Yu (2025), the linear transformer is capable of alleviating this issue, by implicitly discovering the low-dimensional shared structure from the training dataset of multiple domains and automatically applying it as prior information in downstream test tasks. Hence, we anticipate that the linear attention mechanism is capable of discovering the shared structural consistencies, so as to mitigate the deficiency rank issue in small observed data DAG learning problems. In our empirical experiments, we validate these prospectives by showing that: 1) the linear transformer finds a low-dimensional structure in the prior distribution where the ground-truth $A$ is drawn (see Figure 2); and 2) our ADAG is capable of recovering both the correct graph structure and the parameters in $A$, even with a relatively small $n$.

## 4 EXPERIMENTS

In this section we evaluate the performance of our proposed ADAG algorithm, focusing on three key aspects: 1) The attention-mechanism-based kernel map learned by ADAG successfully captures the underlying mechanisms between data observations and their generative processes, which are encoded in a DAG. 2) We extensively assess ADAG's zero-shot DAG inference performance in terms of accuracy and efficiency on synthetic data generated across a wide range of graph settings. For comprehensive evaluation, we also applied our trained ADAG model on test data that comes from different distributions as the training data. 2) We evaluate our ADAG approach on realistic data, showing its practicability in real-world applications. We present the main results that best support our claims in the main paper and provide additional results in Supplementary C.

**Dataset Settings.** We follow standard protocols for generating synthetic graphs and data. The ground-truth DAGs are sampled from Erdős–Rényi (ER) graphs with a degree $k = \{1, 2, 4\}$[2]. We consider graphs of varying sizes with $d = \{5, 10, 20, 50\}$ nodes. Data is then generated using a linear SEM, where the coefficients for each edge are drawn from $U[-2, -0.5] \cup U[0.5, 2]$. We categorize the data into three types, each defined by progressively weaker assumptions: (1) Heterogeneous data, generated following the procedure in (Lu & Gao, 2023). For training, we generate $M$ domains, each with $n$ samples, using the same DAG structure but with varying edge weights. An additional 1000 domains are held out for testing and are not seen during pre-training. (2) Order-consistent data, generated following the procedure in (Chen et al., 2021). Here, each domain contains $n$ samples generated from different DAGs that share the same topological ordering. We again generate $M$ domains for training and 1000 for zero-shot DAG inference. (3) General data, where no assumptions are made about the DAG structure. We randomly generate $M$ training domains, each with $n$ samples drawn from a different random DAG, and generate an additional 1000 samples for zero-shot DAG inference. We provide visualization regarding the heterogeneous data and order-consistent data in Supplementary B.

**Implementation Details.** All experiments are conducted on a single NVIDIA GeForce RTX 5090 GPU using the Adam optimizer for training. Detailed hyperparameter settings—such as the number of layers $nb$, the attention head dimension $k$, batch size, learning rate, total training epochs, and the initialization values for the augmented Lagrangian multipliers $\rho$ and $\lambda$—are provided in Supplementary A. Each experiment is repeated for three times, and we report the average results.

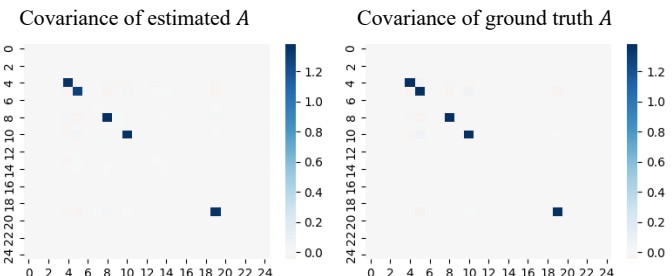

a). Covariance matrices for estimation and ground-truth of weighted adjacency matrix

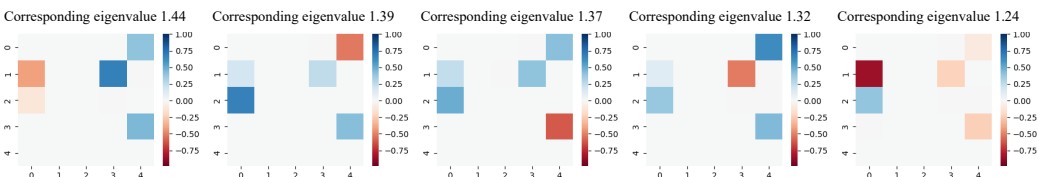

b). Eigenvectors corresponding to the dominant eigenvalues.

Figure 2: Illustration of the learned kernel map on the heterogeneous data generated from an ER1 graph with $d = 5$. Figure (a) shows the covariance matrices of the estimated and ground-truth weighted adjacency matrices. Figure (b) shows the principal components across different domains, by expending adjacency matrix $A^\eta$ as a vector and performing PCA on them. Results shows that all $A^\eta$s are on a dimention-5 space, which are aligned with the ground-truth DAGs.

---

[2]Graphs with degree $k$ have an expected number of edges equal to $kd$.

## 4.1 Generalization of the Nonlinear Kernel Map

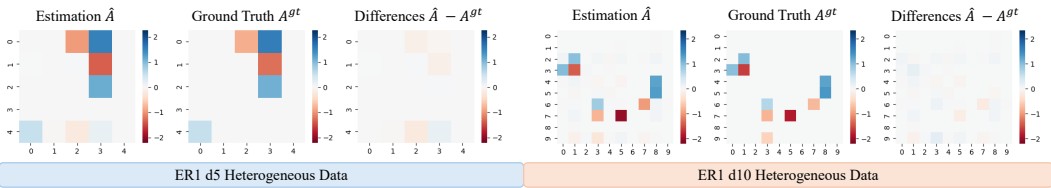

Figure 3: Visualization of the estimation and ground-truth weighted adjacency matrices, along with their difference on the heterogeneous data generated from ER1 graph with $d = 5$ and $d = 10$.

In this section, we show that the attention-mechanism-based kernel map we learned by solving multiple DAG learning problems on training data can generalize well to unseen held-out domains if the DAG structure has appeared in the training set. First, we illustrate that the learned kernel map can identify the common low-dimensional structure across domains. Specifically, we apply the trained kernel map to infer the weighted adjacency matrices $A^{\text{test},\eta} \in \mathbb{R}^{d \times d}$ for 1000 held-out test domains generated from heterogeneous data with $d = 5$. On 1000 test domains, each presented by an $n \times d$ data matrix, our model maps the input tensor of shape $1000 \times n \times d$ to an output of shape $1000 \times d \times d$, corresponding to the estimated weighted adjacency matrices. Then, we treat each weighted adjacency matrix as a size $d^2$ vector, and find the principal component among all 1000 test domains by flattening the output to $1000 \times d^2$ and and computing its covariance matrix. Taking the heterogeneous data scenario for instance, since all domains share the same graph, the dimension among all weighted adjacency matrices should be equal to the number of edges in this hidden graph, and that should be the number of dominant principle components in the above analysis, as observed in Figure 2. Therefore, the dominant diagonal entries (corresponding to the principal components of the weighted adjacency matrixes) align well with those from the ground-truth weighted adjacency matrices. This alignment suggests that the model successfully captures the true manifold in the underlying graphs.

Beyond its ability to identify the common low-dimensional structure, we also show that the learned kernel map is expressive enough to accurately predict the coefficients of the edges in the DAGs. As shown in Figure 3, the weighted adjacency matrices predicted by the learned kernel map closely match the ground-truth weighted adjacency matrices. We observe that this generalization capability hinges on having a sufficient number of domains in the training set. To further investigate this, we conduct an ablation study by varying the number of training domains. As the number of domains increases, the performance of the learned kernel map improves, leading to lower input reconstruction error and reduced relative error between the estimated and ground-truth weighted adjacency matrices. Please refer to Supplementary C.2 for more details.

## 4.2 Zero-Shot DAG Inference

We evaluate the zero-shot DAG inference performance of our proposed ADAG algorithm. Specifically, we apply the learned kernel map, trained on the $M$ training domains, to infer the weighted adjacency matrices for 1000 held-out test domains. Following the common practice in DAG learning (Zheng et al., 2018), a threshold is applied on the inferred weighted adjacency matrices, with a fixed value of 0.3 across all experiments. To assess accuracy, we use the Structural Hamming Distance (SHD), which counts the number of extra, missing, and reversed edges in the inferred DAGs relative to the ground-truth DAGs. Additionally, we report the runtime of the inference process to demonstrate the efficiency of our approach.

We compare the performance of ADAG with two state-of-the-art single-task baselines: NOTEARS (Zheng et al., 2018), DAGMA (Bello et al., 2022); three multi-task baselines: CD-NOD (Zhang et al., 2017), MetaDAG (Lu & Gao, 2023), and MultiDAG (Chen et al., 2021); and two amortized DAG learning baselines: AVICI (Lorch et al., 2022) and FIP (Scetbon et al., 2024). For the single-task methods NOTEARS and DAGMA, we apply each algorithm independently to every domain. For the multi-task methods MetaDAG and MultiDAG, we run the algorithms jointly on the 1000 test domains to simultaneously learn the domain-specific weighted adjacency matrices. For CD-NOD, which is a constraint-based method that returns a DAG skeleton rather than a weighted

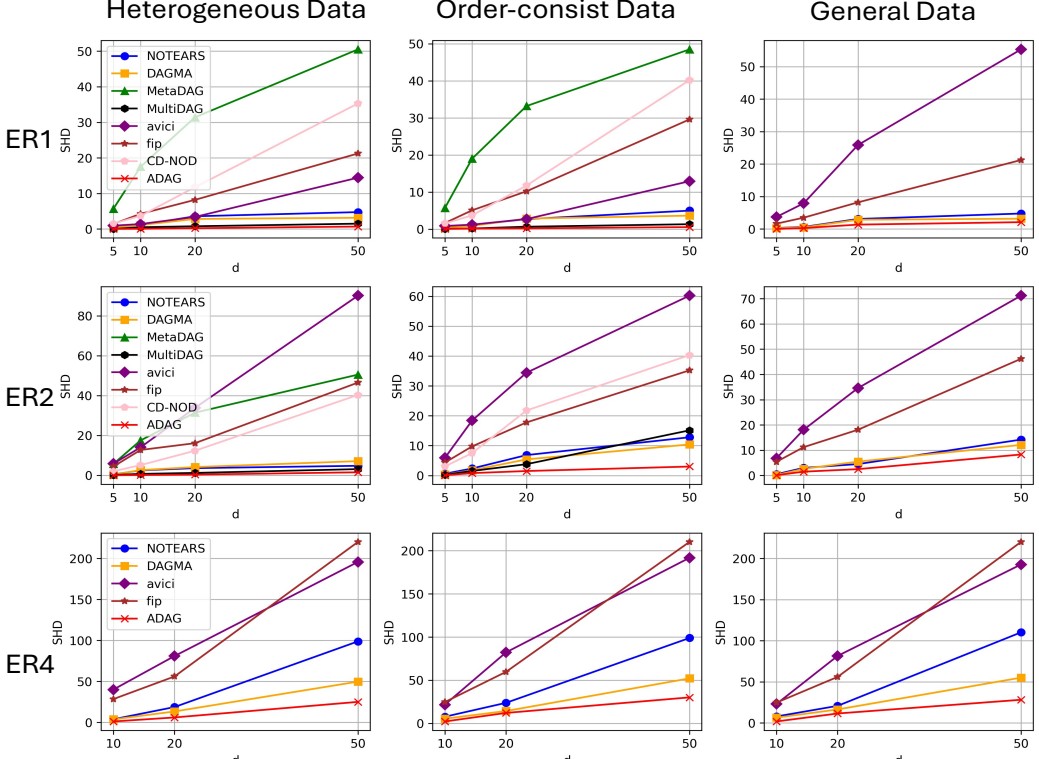

Figure 4: The empirical results of our ADAG approach against state-of-the-art baselines across ER1, ER2 and ER4 graphs on three types of data.

adjacency matrix, we concatenate all observations from the 1000 test domains and use the combined dataset as input. For amortized approaches, we either use a pre-trained model (AVICI) or conduct additional learning after determining the order (FIP). To ensure a fair comparison across all methods, we either extensively tune the hyperparameters or adopt the recommended settings reported in the original papers.

Table 1: Evaluation of the zero-shot DAG inference efficiency performance average over all types of graphs (ER1, ER2, and ER4) and data types (**heterogeneous** data, **order-consistent** data and **general** data) with varying scales.

| d | NOTEARS | DAGMA | MetaDAG | MultiDAG | AVICI | FIP | ADAG |
|---|---|---|---|---|---|---|---|
| 5 | 0.07 | 0.38 | 75.10 | 0.02 | 0.20 | 0.15 | 3e-4 |
| 10 | 0.38 | 0.45 | 213.97 | 00.16 | 0.20 | 0.18 | 4e-4 |
| 20 | 2.15 | 0.70 | 456.53 | 0.45 | 0.20 | 1.05 | 4e-4 |
| 50 | 84.35 | 3.6 | > 5 min | > 5 min | 5.12 | 0.25 | 6e-4 |

We report the DAG learning accuracy performance in Figure 4 and efficiency performance in Table 1. All results are averaged over the 1000 test domains and three independent runs. According to Figure 4 and Table 1, we observe that our ADAG approach achieves the best overall performance in terms of DAG inference accuracy (lowest SHD) and zero-shot inference efficiency, consistently outperforming all state-of-the-art baselines. Moreover, we observe that the advantages of our ADAG approach become more pronounced in challenging cases with large $d$ and in general data without additional assumptions on the DAG structures.

**Evaluation on realistic dataset.** We evaluate our ADAG approach on two realistic datasets, Sachs (Sachs et al., 2005) and Sergio (Dibaeinia & Sinha, 2020). The Sachs dataset contains flow cytometry measurements modeling protein signaling pathways, consisting of 11 continuous variables and 853 observations. We pre-train an ADAG model with $d = 11$ and $n = 100$ on general data, then test it on Sachs using 100 observations randomly sampled from the full dataset. This pro-

cess is repeated 10 times, and we report the mean SHD against several baselines in Table 2. For the Sergio dataset, due to limited computational resources, we do not scale to the full 100-gene E. coli network. Instead, we sample subgraphs with 10 variables per domain to generate a test dataset with 1000 domains. We then apply our ADAG model pre-trained on the $d = 10$ general case and report the results in Table 2. Empirical results demonstrate that pre-trained ADAG models, when trained

Table 2: Empirical Results on Realistic Data Sachs and Sergio

| Methods | Sachs | | | Sergio-Eco.li-subgraphs | |
|---|---|---|---|---|---|
| | SHD($\downarrow$) | # estimated edges | Inference time (s) | SHD($\downarrow$) | Inference time (s) |
| NOTEARS | 15 | 17 | 0.48 | 3.61 | 0.57 |
| DAGMA | 14 | 6 | 0.30 | 3.40 | 0.75 |
| AVICI | 22 | 12 | 0.20 | 6.41 | 0.02 |
| ADAG | **12.45** | 17 | **3e-4** | **2.48** | **3e-4** |

on a sufficient number of domains, achieve strong DAG learning performance on realistic datasets.

Although the true distribution of realistic data is unknown, it is generally assumed to involve non-linear causal relationships between variables, raising the question of how well our ADAG model, pre-trained under linear SEM assumptions, can generalize to data generated from nonlinear SEMs.

**Evaluation on nonlinear synthetic dataset.** We generate nonlinear data using the ground-truth graphs from the held-out test domain and directly evaluate our pre-trained ADAG models on this data under the $ER2$-$d10$ graph setting, with results reported in Table 3. Empirical results show that, even when evaluated on data generated from different types of SEMs, our model retains a certain level of generalizability and achieves performance comparable to baseline DAG learning approaches.

Table 3: The performance on nonlinear data on $ER2$ $d = 10$ settings.

| Methods | Heteogeneous Data | Order-consist Data | General Data |
|---|---|---|---|
| NOTEARS | 0.52 | 0.46 | 0.67 |
| DAGMA | **0.41** | **0.32** | 0.53 |
| AVICI | 2.58 | 4.32 | 6.10 |
| ADAG | 0.42 | 0.35 | **0.45** |

Additionally, we further investigate the capability of our kernel map from ADAG in mitigating the ill-posedness of DAG learning in low-sample regimes. Empirical results demosntrate that ADAG approach can perform accurate DAG inference even when test domains have severely limited data. Such flexibility and adaptability make our model particularly well-suited for real-world applications, where data scarcity is common and collecting additional observations can be costly or infeasible. More numerical results can be find in Appendix.

## 5 CONCLUSION

In this paper, we propose ADAG, a novel attention mechanism-based approach for training a foundation model for DAG learning. The core of our method is a nonlinear kernel mapping that captures the relationship between data observations and their underlying causal structures and mechanisms. By jointly training the model with optimization-based DAG learning approach across multiple domains, ADAG is designed to generalize effectively to test domains with unseen DAGs and mechanisms. Empirically, we demonstrate that the learned kernel map accurately captures the common low-dimensional causal structure and predicts edge coefficients with high precision. Evaluations on benchmark synthetic datasets show that ADAG achieves significant improvements in both DAG learning accuracy and zero-shot inference efficiency. Furthermore, our model exhibits strong robustness in low-sample regimes.

**Limitations and Broader Impact.** Due to computational resource limit, our experiments focus on learning from data generated by linear models with variable size up to $d \leq 50$. It would be beneficial to test the proposed method to even larger variable sizes. Our work takes a meaningful first step toward building generalizable and data-efficient causal discovery systems by introducing a foundation model pre-trained for DAG learning. This has the potential to benefit domains where causal inference is critical but labeled or interventional data are scarce.

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

## A    IMPLEMENTATION DETAILS

In this section, we discuss our implementation in terms of three aspects: attention mechanism model, training procedure, and the augmented Lagrangian optimization.

**Attention Mechanism Model.** We implement our nonlinear kernel map between data observations and weighted adjacency matrices using linear transformers. The key hyperparameters include the number of attention heads $r$, the number of transformer layers $nb$, and the dimension $k$ for the parameters $\mathbf{W}_{1:L}^Q$ and $\mathbf{W}_{1:L}^K$. These parameters are chosen to ensure that the kernel map is expressive enough to generalize to unseen data observations. Specifically, we set $r = 1$ across all settings. For the number of layers, we use $nb = 15$ when $d = 5$ or $d = 10$, and $nb = 20$ when $d = 20$. The dimension $k$ is used to reduce the input observation size $n$, and we typically set $k = \sqrt{n}$. Accordingly, we choose $k = 10$ for $n = 100$ and $n = 50$, and $k = 5$ for $n = 25$.

**Augmented Lagrangian Optimization.** We initialize the Lagrangian multipliers with $\alpha = 0$ and $\rho = 1$. The progress rate is set to $c = \frac{1}{4}$, and the convergence tolerance is $\epsilon = 10^{-5}$. For each value of $\alpha$, we evaluate the acyclicity constraint $\sum_{\eta=1}^{M} h(\hat{A}^{\eta})$. If the constraint does not decrease by a factor of $c$ (i.e., is not reduced to $\frac{1}{4}$ of its previous value), we increase $\rho$ by a factor of 10 and repeat the optimization. If the reduction criterion is met, we update the multiplier as $\alpha \leftarrow \alpha + \rho \sum_{\eta=1}^{M} h(\hat{A}^{\eta})$. The optimization terminates once the constraint value satisfies $\sum_{\eta=1}^{M} h(\hat{A}^{\eta}) < \epsilon$.

**Training Procedures.** We use the Adam optimizer across all settings with a fixed batch size of 100. When $\alpha = 0$ and $\rho = 1$, we train for 5000 epochs with an initial learning rate of $3 \times 10^{-4}$. The learning rate decays by a factor of 0.7 every 1000 steps. For subsequent values of the Lagrangian multiplier, we reduce the number of training epochs to 100 and set the learning rate to $1 \times 10^{-4}$.

## B    DATA VISUALIZATION

We describe the data generation process for both heterogeneous and order-consistent settings. Since both types use the same linear SEM with additive noise to generate observations from ground-truth weighted adjacency matrices, the primary difference lies in the structure of these matrices. Therefore, we illustrate the possible sets of ground-truth adjacency matrices $A^{gt}$ for each setting in Figure 5. As shown in Figure 5(a), the ground-truth weighted adjacency matrices for heterogeneous data share the same DAG structure but differ in their edge weights. According to Figure 5(b), the weighted adjacency matrices for order-consistent data vary in structure but all respect the same underlying topological order.

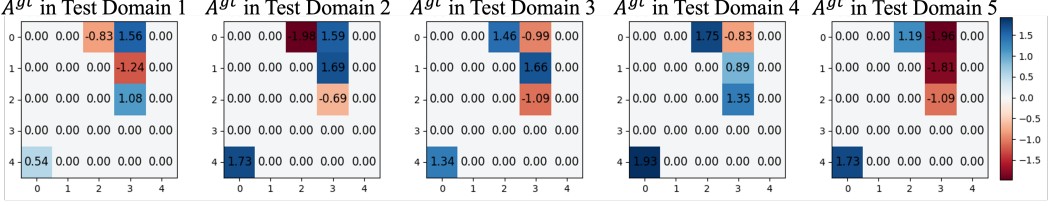

(a) Weighted Adjacency Matrices for Generating Heterogeneous Data

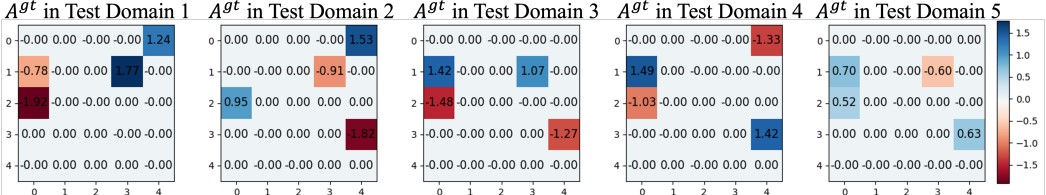

(b) Weighted Adjacency Matrices for Generating Order Consistent Data

Figure 5: Visualization of the Ground Truth Weighted Adjacency Matrices for Heterogeneous Data and Order Consist Data.

## C    DETAILED EMPIRICAL RESULTS

To provide a comprehensive evaluation of our proposed ADAG method, we conduct an ablation study in Section C.2 to examine how the number of training domains influences the generalization ability of the pre-trained model. This study offers empirical insights into the amount of data required for effective pre-training. Furthermore, in Section C.3, we demonstrate that ADAG remains effective under linear SEMs with non-Gaussian noise.

### C.1 LOW-SAMPLE ROBUSTNESS

We further investigate the capability of our kernel map from ADAG in mitigating the ill-posedness of DAG learning in low-sample regimes. To evaluate this, we reduce the number of available observations per domain in both the training and test sets to $n = 25$ and $n = 50$ for the $d = 5$ setting. We then assess the performance of our method and all baselines under these limited-data scenarios and summarize the results in Table 4. While other baselines experience severe performance degradation at $n = 25$ and $n = 50$, our method experiences a relatively modest drop in accuracy, demonstrating greater robustness in the low-sample regime. Additionally, it is of interests to see if our pre-trained foundation model can generalize to test tasks with small samples. To this end, we train ADGA with $n = 100$ observations per domain and apply it to infer the weighted adjacency matrices for test domains with only 25 or 50 observations per domain. To perform the downstream tests, we randomly sample from the available $n = 25$ or 50 observations in each test domain, and augment them with duplicates until the total reaches 100. Then, we use this augmented data as the input for inference in ADAG. As shown in Table 4, models trained with larger $n$ values consistently improve DAG learning accuracy, though at the cost of slightly increased inference time. These results highlight a key advantage of our ADAG approach: its ability to generalize effectively from high-resource to low-resource settings. By pre-training on domains with sufficient data, the model can perform accurate DAG inference even when test domains have severely limited data, simply by leveraging augmentation strategies to align with the pre-training regime. Such flexibility and adaptability make our model particularly well-suited for real-world applications, where data scarcity is common and collecting additional observations can be costly or infeasible.

Table 4: Evaluation of the zero-shot DAG inference performance on ER1 **heterogeneous** data and **order-consistent** data under low-samples regime.

| n | Methods | Heterogeneous Data | | | | Order-consistent Data | | | |
|---|---|---|---|---|---|---|---|---|---|
| | | SHD↓ | $\frac{\|\tilde{A}-A^{gt}\|}{\|A^{gt}\|}$ ↓ | # edges | runtime (s)↓ | SHD↓ | $\frac{\|\tilde{A}-A^{gt}\|}{\|A^{gt}\|}$ ↓ | # edges | runtime (s)↓ |
| | NOTEARS | 0.6590 | 0.2139 | 5.3850 | 0.0757 | 0.5400 | 0.2049 | 4.4020 | 0.0945 |
| | DAGMA | 0.6080 | 0.2028 | 5.4110 | 0.3716 | 0.5440 | 0.2061 | 4.4220 | 0.3591 |
| | MetaDAG | 5.8000 | 1.1018 | 4.7000 | 73.9802 | 5.7000 | 1.0088 | 5.0000 | 72.7758 |
| 50 | CD-NOD | 2.4000 | - | - | 0.7445 | 1.2900 | - | - | 0.6622 |
| | MultiDAG | 0.1110 | 0.1751 | 5.0570 | 0.0114 | 0.5830 | 0.2688 | 4.5430 | 0.4502 |
| | ADAG ($n = 50$) | 0.0550 | 0.1573 | 4.9750 | **0.0004** | 0.1880 | 0.1719 | 4.1320 | **0.0003** |
| | ADAG ($n = 100$) | **0.0540** | **0.1528** | 4.9460 | 0.0006 | **0.0780** | **0.1470** | 4.0260 | 0.0004 |
| | NOTEARS | 1.1870 | 0.2989 | 5.5710 | 0.0710 | 1.5110 | 0.3320 | 5.2160 | 0.0960 |
| | DAGMA | 1.1220 | 0.2852 | 5.5880 | 0.3851 | 1.5590 | 0.3408 | 5.2540 | 0.3657 |
| | MetaDAG | 5.7000 | 1.1016 | 4.7000 | 73.8623 | 5.7000 | 1.0089 | 4.9000 | 72.7697 |
| 25 | CD-NOD | 3.6000 | - | - | 0.3926 | 2.3400 | - | - | 0.3041 |
| | MultiDAG | 0.5610 | 0.2458 | 5.3450 | 0.0097 | 1.0700 | 0.3133 | 4.9180 | 0.3816 |
| | ADAG ($n = 25$) | 0.2710 | 0.2295 | 5.0050 | **0.0004** | 0.6840 | 0.2485 | 4.5200 | **0.0003** |
| | ADAG ($n = 100$) | **0.1250** | **0.1955** | 4.8870 | 0.0006 | **0.1920** | **0.1886** | 4.0820 | 0.0004 |

### C.2 ABLATION ON NUMBER OF DOMAINS

During the pre-training phase, we observe that a sufficiently large number of training domains is necessary to effectively train the nonlinear kernel map, enabling it to produce accurate weighted adjacency matrix predictions for unseen data observations. Hence, we perform an ablation study which varies the number of data domains for training and evaluates the pre-trained models on 1000 test domains data. We set the number of training domains to $M = 0, 500, 1000, 5000, 10000, 15000, 20000, 30000, 40000, 50000, 60000, 70000$, and report the performance in terms of reconstruction loss values and relative errors on the test domains. Figure 6 shows that as the number of training domains increases, both the reconstructed input data observations and the estimated weighted adjacency matrices become closer to the ground truth. When $M = 70,000$, the reconstruction loss (2.5030) and relative error (0.0386) on the test domains are comparable to those on the training domains (reconstruction loss: 2.4980, relative error: 0.0242). Hence, we use $M \geq 70000$ domains for training across all settings.

### C.3 ABLATION STUDY ON VARIOUS TYPES OF NOISE

We also perform ablation study to apply our ADAG on data generated from linear SEMs but with non-Gaussian noise. As shown in Table 5, we generate synthetic data with exponential and Gum-

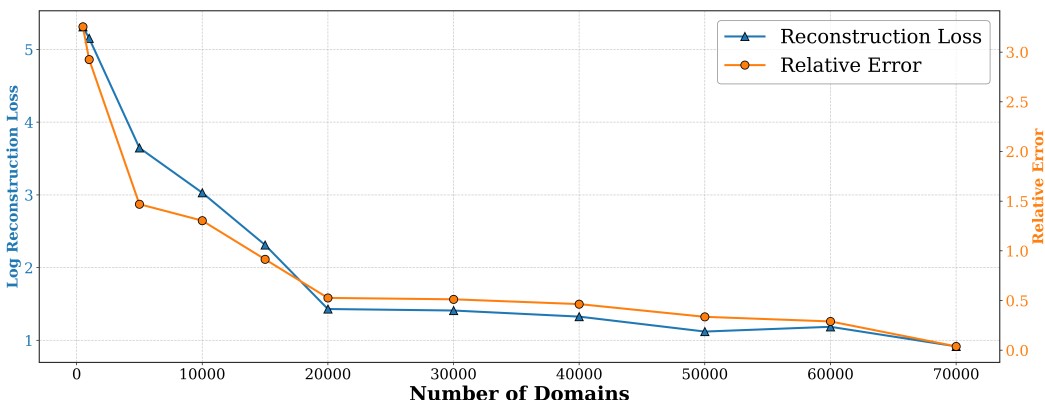

Figure 6: Nonlinear Kernel Map Generalization vs Number of Domains

bel noise distributions, and compare our ADAG method against the state-of-the-art single-task and multi-task DAG learning baselines. The empirical results are consistent with those reported in the

Table 5: Empirical Results of Baselines and our ADAG approach on Linear SEM Data Generated from ER1 d5 Graphs with Various Types of Noise.

| Additive Noise Types | Methods | SHD↓ | # edges | Relative Error ↓ | Runtime(s) ↓ |
|---|---|---|---|---|---|
| Exponential | NOTEARS | $0.5640_{\pm 0.0238}$ | $5.3480_{\pm 0.0243}$ | $0.1620_{\pm 0.0046}$ | 0.0982 |
| | DAGMA | $0.5590_{\pm 0.0152}$ | $5.3900_{\pm 0.0147}$ | $0.1570_{\pm 0.0044}$ | 0.5414 |
| | MetaDAG | $5.5000_{\pm 0.5000}$ | $3.0000_{\pm 1.0000}$ | $0.9489_{\pm 0.0599}$ | 0.2801 |
| | CD-NOD | $1.1500_{\pm 0.1212}$ | - | - | 2.8525 |
| | MultiDAG | $1.3700_{\pm 0.0363}$ | $4.9940_{\pm 0.1301}$ | $0.2755_{\pm 0.0085}$ | 0.0310 |
| | ADAG | $\mathbf{0.1130_{\pm 0.0150}}$ | $4.8970_{\pm 0.0161}$ | $0.2017_{\pm 0.0080}$ | $\mathbf{0.0003}$ |
| Gumbel | NOTEARS | $0.4599_{\pm 0.0572}$ | $5.2969_{\pm 0.0405}$ | $0.1460_{\pm 0.0071}$ | 0.1002 |
| | DAGMA | $0.4639_{\pm 0.0575}$ | $5.3330_{\pm 0.0348}$ | $\mathbf{0.1428_{\pm 0.0053}}$ | 0.5696 |
| | MetaDAG | $4.5000_{\pm 0.5000}$ | $4.0000_{\pm 1.5000}$ | $0.9132_{\pm 0.0470}$ | 0.2488 |
| | CD-NOD | $1.2513_{\pm 0.2029}$ | - | - | 2.6850 |
| | MultiDAG | $1.2261_{\pm 0.0442}$ | $6.0031_{\pm 0.0500}$ | $0.2996_{\pm 0.0007}$ | 0.0278 |
| | ADAG | $\mathbf{0.0570_{\pm 0.0050}}$ | $4.9450_{\pm \mathbf{0.0057}}$ | $0.1442_{\pm 0.0021}$ | $\mathbf{0.0003}$ |

main paper for data with equal-variance Gaussian noise. Our ADAG method achieves optimal performance in terms of DAG inference accuracy (lowest SHD) and zero-shot inference efficiency. Additionally, we report the standard deviation of the expected performance over 1,000 domains across three trials. Compared to all baselines, our method exhibits the smallest standard deviation in both SHD, highlighting the stability and reliability of the trained kernel map.

# D THEORETICAL JUSTIFICATIONS

Intuitively, the optimization problem in our pre-training process can be separated into two sub-problems: (i) learning the estimated adjacency matrix $A$ for each domain from input data $\mathbf{X}$ by minimizing the reconstruction loss $\|\mathbf{X} - A^T\mathbf{X}\|_F^2$ under the acyclicity constraint, which corresponds to the standard DAG learning problem; and (ii) learning the nonlinear maps from the input data $\mathbf{X}$ of each domain to its corresponding weighted adjacency matrix $A$. A well-trained ADAG model requires both sub-problems to be effectively solved.

In the following section, we consider order-consistent data. We first discuss whether the weighted adjacency matrices with the ground-truth DAG structure can be identified for all domains (Section D.1), then examine the identifiability of the parameters in the nonlinear kernel map (Section D.2).

### D.1 GRAPH IDENTIFIABILITY

During the pre-training phase of our proposed ADAG approach, we inherently perform multi-task DAG learning across $M$ domains.

When all $M$ domains have sufficient sample complexity, the identifiability problem reduces to whether the causal graph for each individual domain can be uniquely identified from its corresponding observations. For the linear SEM adopted in our framework, existing identifiability results show that the causal graph is identifiable under the following conditions: (1) the additive noise is non-Gaussian (Shimizu et al., 2006), or (2) the additive noise is Gaussian with equal noise variances (Peters & Bühlmann, 2014). Based on these results, we assert that if the SEMs are linear and the noise satisfies either of these conditions, our method can identify the unique causal graph for each domain. These identifiability results may also be extended to nonlinear SEMs with additive noise, as discussed in Hoyer et al. (2008), Mooij et al. (2009), and Peters et al. (2012).

A more interesting scenario occurs when the observations from some domains are not sufficiently complex to identify a unique DAG. In Chen et al. (2021), it was shown that by minimizing the joint data loss from all $M$ domains as discussed in our Section 3.3, this setting is able to recover the order of non-identifiable graphs if (1) the sample complexity index $\frac{d}{s}\sqrt{\frac{n}{d\log d}\frac{(M')^2}{M}}$ is sufficiently large, (2) the sample size $n$ is also sufficiently large (on the order of $\log M + (p+1)\log d$), and (3) the total domain number $M$ is bounded above by $O(d\log d)$. Here, $d$ is the number of random variables, $n$ is the number of observations in each domain, $p$ is the maximum number of parents in DAGs, $s$ is the size of the support union, and $M'$ is the number of domains with identifiable data among the total $M$ domains. While we anticipate the same identifiability results hold true for our learning problem, we also point out that our ADAG focuses on the small data regime, i.e., $n$ is of a similar size as $d$. Under this circumstance, conditions (1) and (2) may be violated. It suggests a possible relaxation of the theoretical results in Chen et al. (2021) and an improved identifiability property under our foundation model setting. We leave such theoretical investigations to a future work.

### D.2 PARAMETER IDENTIFIABILITY OF $A$

In addition to the capability of identifying the common topological ordering across all domains, ADAG is also capable of identifying the weighted adjacency matrix parameters, i.e., $A$. Under this setting, the learning of parameters can be seen as a discrete version of the learning problem considered in Yu et al. (2024), and one can show that the space in which the values of $A$ are identifiable is the closure of a data-adaptive reproducing kernel Hilbert space (RKHS). In particular, when the common topological order is determined as a permutation $\pi$ over $[1:d] := (1, 2, \cdots, d)$ over all domains, we denote the corresponding connectivity matrix as:

$$[C(\pi)]_{ij} = 1, \text{ if } \pi(i) < \pi(j),$$

$$[C(\pi)]_{ij} = 0, \text{ if } \pi(i) \geq \pi(j).$$

Then, we can rewrite the weighted adjacency matrix $A$ as:

$$A = \tilde{A} \circ C(\pi),$$

where $\tilde{A}_{ij} = 0$ if $[C(\pi)]_{ij} = 0$, and $\tilde{A}_{ij} = A_{ij}$ if $[C(\pi)]_{ij} = 1$. $\circ$ denotes the Hadamard product. One can see that the parameter identifiability problem is equivalent to a learning problem of the $d(d-1)/2$ parameters in $\tilde{A}$. Without loss of generality, we consider $\pi$ to be the identity permutation to simplify the notations. Then, we have the following result:

**Lemma D.1** (Space of Identifiability). *The loss function*

$$\sum_{\eta=1}^{M} ||X_{1:d}^{\eta}(1:n) - (\tilde{A} \circ C(\pi))^{\top} X_{1:d}^{\eta}(1:n)||_F^2 \tag{13}$$

*has a unique minimizer $\tilde{A}$ is the closure of a data-adaptive RKHS $H_G$ with a reproducing kernel $\bar{G}$ determined by the training data:*

$$\bar{G}_{ijk} = [\rho'_j \rho'_k]^{-1} G_{jk}, \text{ if } \pi(i) < \pi(j), \pi(i) < \pi(k), \text{ else } \bar{G}_{ijk} = 0.$$

*Here $\rho'$ is the density of the empirical measure $\rho$ defined by*

$$\rho'_j := \frac{1}{Z} \sum_{\eta=1}^{M} \sum_{s=1}^{n} |X_j^{\eta}(s)|,$$

*with $Z$ being the normalizing constant, and $G$ is defined by*

$$G_{jk} := \sum_{\eta=1}^{M} \sum_{s=1}^{n} X_j^{\eta}(s) X_k^{\eta}(s).$$

*Proof.* The loss function can be expanded as:

$$\sum_{\eta=1}^{M} ||X_{1:d}^{\eta}(1:n) - (\tilde{A} \circ C(\pi))^{\top} X_{1:d}^{\eta}(1:n)||_F^2$$

$$= \sum_{\eta=1}^{M} \sum_{s=1}^{n} ||(\tilde{A} \circ C(\pi))^{\top} X_{1:d}^{\eta}(s)||^2 - 2 \sum_{\eta=1}^{M} \sum_{s=1}^{n} (X_{1:d}^{\eta}(s))^{\top} (\tilde{A}^{\eta} \circ C(\pi))^{\top} X_{1:d}^{\eta}(s) + Const$$

$$= \langle \mathcal{L}_{\bar{G}} \tilde{A}, \tilde{A} \rangle_{L_\rho^2} - 2 \langle \tilde{A}, (\tilde{A})^D \rangle_{L_\rho^2} + Const.$$

$\mathcal{L}_{\bar{G}}$ is an operator mapping from an upper triangular $d \times d$ matrix to another upper triangular $d \times d$ matrix, defined as:

$$(\mathcal{L}_{\bar{G}} \tilde{A})_{ij} = \sum_{\eta=1}^{M} \sum_{s=1}^{n} \sum_{k=i+1}^{d} \tilde{A}_{ik} X_j^{\eta}(s) X_k^{\eta}(s) = \sum_{k=i+1}^{d} \tilde{A}_{ik} \bar{G}_{ijk},$$

and $(\tilde{A})^D$ is an upper triangular $d \times d$ matrix satisfying:

$$\langle \tilde{A}, (\tilde{A})^D \rangle_{L_\rho^2} = \sum_{\eta=1}^{M} \sum_{s=1}^{n} (X_{1:d}^{\eta}(s))^{\top} (\tilde{A} \circ C(\pi))^{\top} X_{1:d}^{\eta}(s).$$

This loss function has a unique minimizer in $\text{Null}(\mathcal{L}_{\bar{G}})^{\perp}$. $\qquad\square$

Intuitively, increasing $M$ and $n$ helps to include more data, which would enhance the invertibility of $\bar{G}$ and enlarge the space of identifiability for $\tilde{A}$. For further discussions on how linear transformer functions enhance the identifiability and solve the inverse linear regression problem, we refer to Yu et al. (2024) and Lu & Yu (2025).

