# OpenReview forum: "Towards a Foundation Model Approach for Causal Graph Learning"
_ICLR.cc/2026/Conference — Submitted to ICLR 2026_

### Official Review · Reviewer_XTxK · 2025-10-15

**Soundness:** 2
**Presentation:** 3
**Contribution:** 2
**Rating:** 2
**Confidence:** 4

**Summary:**

This paper introduces Attention-DAG (ADAG), a transformer-based model designed for zero-shot causal discovery. The model is pre-trained across numerous domains to learn a direct mapping from observational data to a linear Structural Equation Model (SEM), including both the graph structure and its parameters. The key idea is to amortize the cost of discovery, enabling extremely fast inference on new, unseen tasks.

**Strengths:**

The paper's ambition to create a foundation model for unsupervised causal discovery is timely and novel. The proposed ADAG architecture is a non-trivial application of transformers to this inverse problem, and the resulting zero-shot inference speed is empirically impressive on the tested benchmarks.

**Weaknesses:**

The paper's central claims are undermined by severe methodological limitations that challenge its viability as a true "foundation model" for causal discovery.
1. The model's foundation is critically narrow, as it is pre-trained exclusively on data generated from linear SEMs. Real-world causal mechanisms are complex, diverse, and often highly nonlinear. By only learning to recognize linear relationships, the model is theoretically unprepared to handle unseen, novel causal functions. While the model can be extended to handle nonlinear data, this capability should not be conflated with the far more difficult challenge of accurately recovering the structure with diverse and unknown nonlinear causal functions that actually generated the data.
2. The paper's framing as a "foundation model" is misleading due to its choice of training objective.ADAG is optimized using a data-reconstruction loss, which is the exact same objective used by single-task, optimization-based methods like NOTEARS. This forces the model to solve the same problem: find a DAG that best fits the linear data-generating process. The underlying causal function depends on the selected loss, which currently can not learn multiple causal functions at the same time. Hence, it does not learn the more general task of mapping data to its corresponding abstract structure.
3. Because the objective is data-fitting, the pre-training process is not learning a general causal discovery skill. Instead, it is simply moving the computational cost of traditional structure learning into a pre-training phase. The model becomes a fast, amortized solver for one specific problem (linear DAG fitting under similar data scales and structures), but it does not learn a fundamentally more general capability.
4. A true foundation model should generalize across a vast diversity of tasks. However, ADAG's methodology suggests that to handle different causal functions (nonlinear, heterogeneous), node scales, and graph structures, one would need to pre-train an entirely new "foundation model" for each specific class of problems. This defeats the purpose and promise of a single, general-purpose foundation model.

**Questions:**

1. Given that the model is trained exclusively on linear SEMs, why should the community trust its output on real-world datasets where the functional forms are unknown, not linear, and shares diverse causal functions? How does this not represent a critical failure of generalization, which is the primary promise of a foundation model?
2. Your training objective is identical to that of NOTEARS—minimizing data reconstruction error under an acyclicity constraint. Why did you not opt for a direct structural loss (e.g., supervising the model to match the ground-truth adjacency matrix)? Doesn't using a data-fit loss simply force the model to become an "amortized NOTEARS," rather than learning the more abstract and generalizable task of mapping data characteristics to a graph structure?
3. The very concept of a foundation model is built on generalization from massive, diverse data. Your approach seems to require a new "foundation model" for each class of causal functions (linear, polynomial, etc.) and potentially for different graph scales. How do you reconcile this with the claim of building a generalizable foundation model for causal discovery?

---

> ### Author Response · Authors · 2025-12-02
> **Responses to reviewer XTxK (part 1)**
>
> We thank the reviewer’s time and efforts. We are concerned that the reviewer's analysis of our training procedure is rooted in a fundamental **misunderstanding of the ADAG methodology**, which has resulted in an inaccurate assessment of the model's capabilities and generality. We have provided comprehensive responses in the subsequent pages that fully address every weakness and question raised, hopefully to help clarify the **robustness of our approach** and the **promising potential of our foundation model for causal discovery**.
>
> **Justification of Linear SEM Settings.** We appreciate the reviewer's concern regarding our choice of a linear SEM setting. We assert that this decision is both **methodologically sound** and a **strategic necessity** for this foundational work.
> 1. **Alignment with Established Practice and Utility**
>      - **Prevalence in Literature**: Linear SEMs are a **well-established and widely accepted paradigm** in the causal discovery literature [1, 2, 3] and continue to be a highly active area of research.
>      - **Proven Real-World Success**: The utility of this approach is validated by recent work, such as Dong et al. (2023) [4], which demonstrates the effectiveness of linear SEMs in addressing real-world problems across diverse domains. A prime example is **EEG/MEG source separation**, which is routinely and successfully modeled using linear mapping functions.
> 2. **Foundational Scope and Strategic Focus**
>      - **Initial Foundational Step**: This work is purposefully designed as our initial step toward building a **pre-trained foundation model** for causal discovery. Our primary goal is to **evaluate the feasibility of learning a nonlinear mapping from data observations to weighted adjacency matrices**.
>      - **Scope Limitation**: To isolate and focus on this core contribution, we strategically **restrict our study to linear SEMs**. This crucial choice allows us to avoid the substantial increase in optimization complexity and the challenges associated with **parameter unidentifiability (besides the graph identifiability)** that are inherent in general nonlinear cases.
> 3. **Demonstrated Generalization**
> Finally, the resulting foundation model, while trained on linear data, is **not as limited as it may appear**. By utilizing the ability to train across vast numbers of domains, our model spans a wide range of DAG structures. This comprehensive training enables it to achieve a **reasonable degree of generalization capability even to unseen nonlinear SEM cases** (as detailed in Table 3).
> While we acknowledge that our current model is not a universal solution for all causal discovery scenarios, it clearly demonstrates **meaningful generalization capabilities** and represents a **promising step** toward building a powerful DAG learning foundation model.
>
> > [1] Zheng, Xun, et al. "Dags with no tears: Continuous optimization for structure learning." Advances in neural information processing systems 31 (2018).
>
> > [2] Park, Gunwoong. "Identifiability of additive noise models using conditional variances." Journal of Machine Learning Research 21.75 (2020): 1-34.
>
> > [3] Dai, Haoyue, Peter Spirtes, and Kun Zhang. "Independence testing-based approach to causal discovery under measurement error and linear non-gaussian models." Advances in Neural Information Processing Systems 35 (2022): 27524-27536.
>
> > [4] Dong, Xinshuai, et al. "A versatile causal discovery framework to allow causally-related hidden variables." arXiv preprint arXiv:2312.11001 (2023).

---

> ### Author Response · Authors · 2025-12-02
> **Responses to reviewer XTxK (part 2)**
>
> **ADAG Training Process.** In response to weakness 2 and question 2, we are concerned that the reviewer's critique regarding "amortized NOTEARS" and the subsequent questions (Weakness 2 and Question 2) are based on a **fundamental misinterpretation of the ADAG methodology**.
>
> ADAG vs. "Amortized NOTEARS":
> - **Our Model**: As made explicit in **Figure 1** and **Equation (4)**, ADAG directly learns a **nonlinear mapping** from domain-specific observations $X^\eta$ to weighted adjacency matrices $A^\eta$, with the objective of learning the shared mapping parameter $\Theta$.
> - **Methodological Separation**: As detailed in lines 801–806, our optimization naturally separates into two distinct and crucial components:
>      1. **Estimating $A^\eta$** for each domain (a single-task DAG learning problem involving data fitting and acyclicity).
>      2. **Learning the nonlinear mapping** from the collection of pairs $(X^\eta, A^\eta)_{\eta=1}^M$ (the core contribution of ADAG).
> - **Reviewer's Focus**: The reviewer's interpretation is limited **exclusively to component (1)** and **ignores the central contribution of component (2)**, which is the **essence of ADAG**—a predictive model for causal graphs.
>
> Core Contribution: Mapping Data to Graph Structure
> We find it highly concerning that Question 2 asks why we do not learn "**the more abstract and generalizable task of mapping data characteristics to a graph structure.**" We must emphasize that this mapping is exactly what our ADAG framework accomplishes. This indicates a serious misreading of the method's objective.
>
> Innovation of the Unsupervised Framework: Regarding the reviewer’s suggestion for a supervised training objective based on relative error:
> - This is indeed a reasonable **supervised objective**. However, as clearly stated in the Abstract, our work introduces the first unsupervised amortized learning framework for causal graph estimation.
> - Our unsupervised formulation, based on a **data reconstruction loss**, fully exploits the expressive power of linear SEMs **without requiring ground-truth graphs or known causal mechanisms**. As pointed out by reviewer qbVU: (unsupervised formulation) makes the method highly practical, as ground-truth causal graphs are almost impossible to obtain in the real world.
> - This distinction—avoiding the need for ground-truth information, is a **central innovation** of our approach, significantly **reducing data demands** and increasing applicability compared to existing amortized methods that rely on supervised knowledge.
>
> **we must point out that the reviewer has fundamentally misunderstood the methodology of our paper. As made explicit in Figure 1 and Eq. (4), ADAG is not performing “amortized NOTEARS,” which appears to be an assumption constructed entirely by the reviewer. Our method maps domain-specific observations $X^\eta$ to weighted adjacency matrices $A^\eta$, and the goal is to learn the shared parameter $\Theta$ governing this mapping. As stated in lines 801–806, the optimization naturally separates into two components: (1) estimating $A^\eta$ for each domain via data fitting under the acyclicity constraint and (2) learning the nonlinear mapping from pairs $(X^\eta, A^\eta)_{\eta=1}^M$. The reviewer’s interpretation is limited exclusively to (1) and ignores the central contribution of (2), which is the essence of ADAG. Moreover, we find it highly concerning that Question 2 asks why we do not learn “the more abstract and generalizable task of mapping data characteristics to a graph structure,” which is exactly what our framework accomplishes. This indicates a serious misreading of the method. Regarding the reviewer’s suggestion to use supervised training based on the relative error between ground-truth and estimated adjacency matrices, we note that this is indeed a reasonable *supervised* objective. However, as clearly stated in the abstract, our work introduces the first unsupervised amortized learning framework for causal graph estimation. Unlike existing amortized approaches that depend on ground-truth graphs or known causal mechanisms, our unsupervised formulation, based on data reconstruction loss, fully exploits the expressive power of linear SEMs without requiring such information, significantly reducing data demands. This distinction is a central innovation of our approach.**

---

> ### Author Response · Authors · 2025-12-02
> **Responses to reviewer XTxK (part 3)**
>
> **Training Complexity and Runtime.** In response to **weakness 3**, we would like to emphasize that the amortized-learning paradigm is exactly how modern foundation models, including large language models, achieve scalability and low inference cost. Training requires upfront resources, but once trained the model generalizes and can be applied to new datasets without retraining. A model trained for small-to-moderate variable sizes also enables efficient inference of local structures on variable subsets, which can be composed into a global graph. Determining whether a viable kernel mapping exists between observations and weighted adjacency matrices is the critical first step for scaling, which is why our experiments focus on variables up to $d=50$. For completeness, we provide both a complexity analysis and measured training times. Our architecture, which uses linear transformer blocks, has computational complexity approximately $\mathcal{O}(Mdn^2)$. In practice both the number of domains $M$ and the number of variables $d$ drive training time; for larger $d$, the required number of domains $M$ to cover the space of graphs typically increases, sometimes substantially. Empirically, training on general synthetic data ranges from about 13 minutes to 40 hours for cases with $d=5$ to $d=50$.
>
>
> **Generality of the Foundation Model.** In response to Weakness 4 and Question 3, we emphasize that it is **not necessary to train separate foundation models** for different classes of causal functions (linear/nonlinear) or graph sizes. This critique misunderstands the goal of a general foundation model. Our ultimate goal is to pre-train a single, **sufficiently general foundation model** on a large number of domains with a relatively large variable size, $d$. Once trained, this single model can be applied universally:
> **Variable Size Adaptation**: The model can be applied to **any dataset with up to $d$ variables**. If the number of variables in a test domain, $d^{\text{test}}$, is less than $d$, we can simply **pad the remaining $d - d^{\text{test}}$ observations with zeros** and still apply the trained foundation model to accurately infer the causal graph.
> **Function Generalization**: The model achieves **very high accuracy on linear data** and retains a **reasonable level of accuracy on nonlinear data**, demonstrating significant built-in generalization capability.
> This work represents our **first step** toward building a truly pre-trained foundation model for causal discovery. We acknowledge that foundation models, by nature, require **multiple iterations of training and refinement**, a process that takes time. Our goal here is to **demonstrate the feasibility of pre-training a sufficiently general model**. This core achievement establishes a reliable starting point that can be scaled, optimized, and improved through future iterations, paving the way for a powerful, universal tool in causal discovery.

---

### Official Review · Reviewer_qbVU · 2025-10-23

**Soundness:** 1
**Presentation:** 2
**Contribution:** 2
**Rating:** 4
**Confidence:** 3

**Summary:**

This paper proposes a novel causal graph learning method named Attention-DAG (ADAG) , whose core idea is to introduce the concept of a 'Foundation Model' to address the challenges faced by traditional causal discovery methods, such as high computational cost (super-exponential space) and poor performance in small-sample regimes. ADAG utilizes a Transformer-based attention mechanism to learn a nonlinear 'kernel map' , which can directly convert observational data into the corresponding weighted adjacency matrices (i.e., graph structure and parameters) of linear Structural Equation Models (SEMs). This model, through joint pre-training across multiple domains (tasks) , learns a shared low-dimensional structural prior in an unsupervised manner (using a data reconstruction loss and an acyclicity constraint ). This pre-training enables ADAG to capture commonalities across tasks , effectively mitigating the ill-posedness problem in small-sample learning. Once trained, ADAG can perform efficient 'zero-shot inference'.

**Strengths:**

Once the model is trained, its inference speed during the testing phase is extremely fast. As shown in Table 1, ADAG's inference time is far lower than that of baselines (like NOTEARS, DAGMA) which require re-optimization for each task. This holds immense practical value in scenarios requiring the rapid processing of numerous different causal tasks.

By pre-training on multiple tasks to learn a shared low-dimensional prior, ADAG is designed to solve the small-sample problem. Experiments fully demonstrate its robustness in small-sample contexts, significantly outperforming other baseline methods.

The method is trained using a data reconstruction loss and does not require ground-truth DAG structures as labels. This makes the method highly practical, as ground-truth causal graphs are almost impossible to obtain in the real world.

**Weaknesses:**

The experimental results of this paper are intriguing. The authors claim their method learns a "low-dimensional structural prior". Yet, in the "General Data" experimental setting, the DAG for each domain is randomly generated. The paper fails to clearly explain what "shared low-dimensional structural prior" the model could possibly be learning in this scenario. If the graphs are random, no shared structure should theoretically exist. The model's surprisingly strong performance under this setting (Figure 4) is puzzling. This raises the suspicion that what is being learned is not a structural prior, but rather a general "algorithmic prior"—akin to a universal solver for this class of linear problems.



Furthermore, even if we accept the algorithm's validity in principle, its practical feasibility is questionable. As shown in Figure 6, in nonlinear cases, joint training on at least 20,000 domains is required to achieve a converged result. Leaving aside the computational cost, obtaining such a vast number of training domains that share a common structure is highly unrealistic in most real-world applications. The paper's two "real dataset" experiments further underscore this, as the models were trained on synthetic data, not real-world domains. Consequently, I hold significant skepticism regarding both the practical reasonableness and the overall contribution of the proposed method.

**Questions:**

The paper's mathematical formulation (e.g., Equation 9) appears to use a reconstruction loss consistent with a standard Gaussian noise assumption, but it does not explicitly incorporate identifiability conditions such as 'equal variances' or 'non-Gaussian noise'. In a linear Gaussian SEM, the model can only identify a Markov Equivalence Class. The paper does not clearly explain how ADAG selects a specific DAG from within this equivalence class.

In all experiments, the authors use a fixed threshold of 0.3 to extract the final graph structure from the weighted adjacency matrix. This is a highly sensitive hyperparameter. Could the model be designed to automatically learn an appropriate threshold? Alternatively, a threshold-independent metric (e.g., AUC-PR for edge prediction) should be reported during evaluation to demonstrate that the model's performance is not sensitive to this choice.

The model's input tokenization method ($X^{n}(1:n)\in\mathbb{R}^{d\times n}$) is unique, treating each variable (across n samples) as a single token. This differs from the more common approach in Transformers, which typically treats each sample (across d variables) as a token. What is the justification for this choice?

The initial training time of the proposed method, and how it scales with the dataset size, is also a critical metric. It is insufficient to only report the inference runtime; this information should be supplemented.

---

> ### Author Response · Authors · 2025-12-02
> **Responses to reviewer qbVU (part 1)**
>
> We thank the reviewer for their insightful comments, especially for recognizing the application value of providing zero-shot DAG learning in an unsupervised manner.
>
> Our response:
>
> **Clarification regarding how ADAG works on general data.** We note that ADAG discovers the underlying structure of $A$ across different domains, which of course depends on the underlying domain setting. As demonstrated in Figure 2, when there is a low-dimensional structure, as in the cases of heterogeneous data and order-consistent data, ADAG has successfully discovered it. On the other hand, when there is no low-dimensional structure in $A$, as in the general data case, ADAG learns the mapping from $X$ to $A$, as long as it has been exposed to various causal graph structures during training. That has been said, ADAG is capable of discovering the correct structure of $A$, but it is **not restricted to** the scenarios with a low-dimensional structural prior. From that point of view, ADAG can be seen as learning a combination of structural prior together with algorithmic prior. Intuitively, it learns the nonlinear solver of the following nonlinear optimization problem:
>
> $$
> A^* = \arg\min_A  ||X - AX||_{F}^{2} + \lambda_1||A||^{2} + \lambda_2 h(A)
> $$
>
> where $||A||$ is the corresponding norm determined by the prior from all $A$ in the training domains, aka, the structural prior, and hence we can specify it as $||A||_{\Sigma(A)}$. And $h(A)$  denotes the acyclic constraint, aka, the algorithmic prior. If ADAG only learns the algorithmic prior, it should be equivalent to the baseline model, NOTEARS, since the later solves the optimization problem using exactly the same algorithm and acyclic constraint.
>
> **Identifiability Conditions.** We note that the identifiable conditions the reviewer refers to are already **explicitly stated** in lines 817–819. In the same section, we also clearly specify that for each single-task DAG learning problem, we **assume the underlying causal graph is identifiable**. Under this standard assumption, the issue of selecting a single DAG from within an equivalence class becomes moot, as the underlying DAG is **uniquely determined** by the observational data within that specific domain.
>
> **Threshold-independent Metric**. We adopt a fixed threshold of 0.3, following the standard protocol in Zheng et al. (2018). This choice is reasonable because we only treat edges with absolute weight greater than 0.5 as causal, so a 0.3 threshold ensures stability and consistency in graph extraction. We appreciate the reviewer’s suggestion and agree that an adaptive threshold could indeed be learned. For example, one could introduce the threshold as **a trainable parameter** and apply a learnable mask to the estimated $A$, **optimizing it jointly during training**. Regarding the reviewer’s recommendation to use threshold-insensitive metrics such as AUC-PR for edge prediction, we note that we already employ a metric that is **fully threshold-independent** and more directly aligned with our claims: the relative error between the estimated $A$ and the ground-truth $A$. This metric evaluates both structural accuracy and the precision of the learned edge coefficients (causal mechanism weights). We report these results in Tables 4 and 5. We choose SHD as our main empirical metric to allow a broader comparison with standard baselines, since many causal discovery methods do not output weighted coefficients and thus cannot be evaluated using relative error. SHD allows us to benchmark ADAG's structural accuracy against the wider literature.
>
>
> **Tokenization Approach.** We appreciate the reviewer’s insightful question. Treating each variable as a token is one of the keys and architecture novelties in ADAG. In DAG learning, the goal is to infer the relationship between different variables, which is consistent with inferring the nonlocal weight between tokens in the attention mechanism. In fact, taking the data observation for one domain as an input matrix $X:=X^\eta(1:n)\in \mathbb{R}^{d\times n}$, the attention block writes:
> $output=W^P \sigma(X W^Q (W^K)^\top X^\top) X$
> Comparing this formulation with our data model in equation (1), one can see that when taking $output=X-E$ and $A=W^P \sigma(X W^Q (W^K)^\top X^\top)$, the attention block is consistent with the linear structural equation model, and the attention matrix provides an estimator of the weighted adjacency matrix A when training it with the standard data loss $||X-ADAG(X) X||^2_F$.

---

> ### Author Response · Authors · 2025-12-02
> **Responses to reviewer qbVU (part 2)**
>
> **Training Complexity and Runtime.** Regarding training cost, we emphasize that the amortized-learning paradigm is exactly how modern foundation models, including large language models, achieve scalability and low inference cost. Training requires upfront resources, but once trained the model generalizes and can be applied to new datasets for **zero-shot inference** without requiring any further retraining. A foundation model trained for small-to-moderate variable sizes (up to $d=50$ in our experiments) still enables highly efficient inference of **local structures** on variable subsets, which can then be composed to form a global graph for larger systems.. The critical first step toward full scalability is confirming that a **viable kernel mapping exists** between observational data and weighted adjacency matrices, which justifies the focus on variables up to $d=50$ in the current work. For completeness, we provide both a complexity analysis and measured training times. Our architecture, which uses linear transformer blocks, has computational complexity approximately $\mathcal{O}(Mdn^2)$. In practice both the number of domains $M$ and the number of variables $d$ drive training time; for larger $d$, the required number of domains $M$ to cover the space of graphs typically increases, sometimes substantially. Empirically, training on general synthetic data ranges from about 13 minutes for $d=5$ and 40 hours for $d=50$.

---

### Official Review · Reviewer_5QJS · 2025-10-26

**Soundness:** 3
**Presentation:** 2
**Contribution:** 2
**Rating:** 4
**Confidence:** 4

**Summary:**

This paper proposes an attention-driven DAG learning framework, ADAG, which aims to build a generalizable basic model for causal discovery. The model maps observation data to a weighted adjacency matrix by pre-training a nonlinear kernel mapping, thereby jointly learning DAG structure and causal mechanisms in multiple tasks. ADAG supports zero-shot inference, eliminating the need for additional training on unseen tasks, and performs well in data-scarce and small-sample scenarios. Experiments show that ADAG outperforms existing single-task, multi-task, and amortized DAG learning methods on both synthetic data and real data, especially in terms of inference efficiency and accuracy.

**Strengths:**

This paper proposes a pre-trained foundational model for unsupervised DAG learning, breaking through the limitations of traditional single-task DAG learning and addressing the key challenges of cross-domain generalization and small-sample adaptation. It employs an attention mechanism to construct a nonlinear kernel mapping, leveraging the Transformer's ability to model complex dependencies while balancing computational efficiency through linear attention blocks. By incorporating acyclic constraints through an enhanced Lagrangian method and optimizing the data reconstruction loss, the paper theoretically guarantees the validity of the model output and provides a theoretical analysis of parameter identifiability.

The dataset covers synthetic data, real biological datasets, and nonlinear data. The comparison baseline is comprehensive: eight state-of-the-art methods across three categories: single-task, multi-task, and amortized. Fairness is ensured through hyperparameter tuning and the recommended settings in the original paper. Key metrics are outstanding, with significant accuracy advantages in small-sample scenarios.

The application value is clear. To address the problems of data scarcity (such as the difficulty in obtaining biological experimental samples) and variable tasks in real scenarios, ADAG's pre-training paradigm can be directly transferred to downstream tasks without retraining, providing practical tools for the implementation of causal reasoning in medical, biological and other fields.

**Weaknesses:**

Incomplete nonlinear SEM adaptation: While the paper mentions that ADAG can be extended to nonlinear SEM, the final Training Objective stage still uses a linear form. It does not propose a complete architectural design for nonlinear SEM, nor does it compare specialized nonlinear DAG learning methods. Its adaptability to nonlinear scenarios lacks systematic verification. Excluding pre-training on data from multiple systems, it appears to simply replace the linear optimization in Notears with Attention.

Training Optimization Objective: Although Notears obtains an adjacency matrix through reconstruction and constraints, it does not claim to be a causal graph. In ADAG, does the A obtained by reconstructing the observed data truly represent a "causal graph" of causal relationships? This point is not discussed alongside identifiability.

Generalizability of the "pre-trained model": The paper does not provide a detailed explanation of the training data construction process for the pre-trained model. Furthermore, the performance of the pre-trained model is likely to be highly dependent on the distribution and diversity of the pre-training data. Performance may decline if the test data differs significantly from the training distribution, a point that is not fully discussed. Appendix C.2 shows that model performance stabilizes when M ≥ 70,000 training domains. However, the paper fails to analyze the model's degradation when the number of domains is insufficient (e.g., M < 10,000), nor does it explore the impact of domain distribution differences on zero-shot generalization. This results in insufficient support for the model's practicality in real-world scenarios with limited training data.

**Questions:**

Is ADAG a universal model, meaning it can be adapted to any test data without any modifications or fine-tuning? How is its training data constructed? Does it support mixed data types?

For nonlinear SEM, how should the nonlinear transformation layer of the input data be designed to optimally adapt to the attention kernel mapping? Do the acyclicity constraints or loss functions need to be adjusted to accommodate nonlinear data generation mechanisms? Can specific nonlinear DAG learning methods be compared to more comprehensively validate ADAG's advantages in nonlinear scenarios?

Can the paper provide in-depth analysis of the physical meaning of attention weights—for example, the differences in the contributions of different attention heads and layers to the DAG structure (e.g., key causal edges)? Can visualizations (e.g., attention heatmaps) demonstrate how the model "focuses" on key variables to infer causal relationships?

---

> ### Author Response · Authors · 2025-12-02
> **Responses to reviewer 5QJS (part 1)**
>
> We thank the reviewer for their insightful comments, especially for recognizing the application value of providing zero-shot DAG learning and the comprehensive evaluation in our empirical experiments.
>
> Our response:
>
> **Linear Model Limitation.** The focus on **linear SEMs**, and the limitation this imposes, is explicitly acknowledged in our paper, particularly within the Limitations section. This choice is deliberate and well-justified: linear SEMs are highly **prevalent and remain an active area of research** in the causal discovery literature $[1, 2, 3]$. Furthermore, the practical utility of this approach is validated by recent successes $[4]$ and real-world problems like **EEG/MEG source separation**, which is routinely modeled using linear mapping functions. While our primary contribution is the foundation model for linear SEMs, we also demonstrated its **extensibility to nonlinear settings** through a dedicated discussion and a generalization ablation study. This study empirically confirms that the current linear model exhibits a **reasonable degree of robustness** on nonlinear data. We maintain that a full nonlinear formulation of ADAG and its associated learning algorithm is **beyond the defined scope of this foundational work**. We view nonlinear ADAG as a promising and natural direction for future extension.
>
>
> > [1] Zheng, Xun, et al. "Dags with no tears: Continuous optimization for structure learning." Advances in neural information processing systems 31 (2018).
>
> > [2] Park, Gunwoong. "Identifiability of additive noise models using conditional variances." Journal of Machine Learning Research 21.75 (2020): 1-34.
>
> > [3] Dai, Haoyue, Peter Spirtes, and Kun Zhang. "Independence testing-based approach to causal discovery under measurement error and linear non-gaussian models." Advances in Neural Information Processing Systems 35 (2022): 27524-27536.
>
> > [4] Dong, Xinshuai, et al. "A versatile causal discovery framework to allow causally-related hidden variables." arXiv preprint arXiv:2312.11001 (2023).
>
> **Novelty of our ADAG.** The assertion that ADAG is a simple modification of NOTEARS—specifically by replacing the linear causal function with an attention mechanism—misses the **fundamental novelty of our approach**. (Nonlinear) **NOTEARS** is a **single-task method** where a neural network only models the causal mechanisms to reconstruct the data $X$; the resulting causal graph is then inferred with extra steps from the network's parameters. This process yields an adjacency matrix $A$ for a specific dataset. In contrast, our **ADAG model** is designed to learn the **direct, nonlinear mapping from the data $X$ to the weighted adjacency matrix $A$ itself**, outputting $A$ directly. Our contribution lies in training a **predictive model** capable of generalizing to new data. If one were merely to substitute the causal function in NOTEARS with an attention module, the result would remain standard multi-domain DAG estimation: one would obtain a set of adjacency matrices across domains, but **not a trained model capable of predicting the adjacency matrix $A$ from novel data $X$**. This distinction—**learning $A$ as a direct function of $X$ versus inferring $A$ from network parameters**—is central to the purpose, design, and contribution of ADAG as a **foundation model for causal discovery**.
>
> **Correctness of the Causality.** The reviewer correctly notes the inherent challenge in proving the recovery of the true causal graph from observational data in general. Our work adheres to the **standard and established practice** in causal discovery: we **establish the conditions** under which a unique graph is **identifiable** from observational data, thus permitting its interpretation as the underlying causal structure. Our **theoretical justification (Appendix D)** provides the necessary intuition and formal conditions for ADAG to correctly learn the nonlinear mapping between data observations and weighted adjacency matrices.
> Since our training framework modularly decomposes into two parts: **Single-Task DAG Learning (within each domain)**, and **Learning the Nonlinear Mapping (across domains)**. We specify assumptions on the data distribution that **guarantee the identifiability of the linear SEM** in each individual domain. Under these established conditions, if the domain-specific adjacency matrix $A^\eta$ is correctly estimated for a **sufficiently large number of training domains**, the overall nonlinear model can **accurately learn the mapping from data to causal graphs**, while simultaneously maintaining **robustness to a small number of misestimated domains**. This provides the necessary theoretical grounding for the correctness of ADAG.

---

> ### Author Response · Authors · 2025-12-02
> **Responses to reviewer 5QJS**
>
> **Training Domain Data.** We strategically evaluate ADAG on three distinct types of data with progressively increasing difficulty in learning the nonlinear mapping from data to causal graphs.
> - **Heterogeneous Data:** The mapping problem is simpler, as the space of possible causal graphs is restricted to a **single, common DAG** across domains.
> - **Order-Consistent Data**: The complexity is moderate, as the causal graphs are restricted to a **family of DAGs sharing a common causal order**.
> - **General Data**: The most challenging class, where **no structural restrictions are imposed**, allowing the causal graphs to vary freely across domains.
>
> For each class, we ensure a fair evaluation by generating training and testing domains from weighted adjacency matrices that adhere to the corresponding structural assumptions. Crucially, while the **prior distribution over graph structures** $p(\mathcal{G})$ **is consistent** between training and testing, we guarantee that the **specific weighted adjacency matrices are distinct**, preventing any test-domain matrix from appearing during training. We concur that exposing the model to a wider and **sufficiently rich variety of causal graph structures** during training is essential for generalization in the 'General Data' setting. Our **ultimate goal** is to train a **foundation model** on this general data, utilizing the virtually unlimited quantities of easily generated **synthetic linear-SEM data** to span the space of possible DAGs. This approach is designed to enable reliable **zero-shot inference** on entirely unseen domains.
>
> We operate under the assumption that a **sufficiently large number of training domains is available**, consistent with the resource model for training foundation models. Therefore, we do not explore scenarios with severely limited training domains, though we anticipate a natural degradation in ADAG's DAG learning accuracy in such constrained settings. Finally, our foundation models currently handle **continuous data generated from linear SEMs** and do not incorporate mixed data types.
>
> **Intuition regarding the attention-mechanism.**  As shown in Figure 1 and equations (5-7), our ADAG model consists of multiple linear transformer layers followed by a final kernel layer. The linear transformer layers identify complex correlations among variables and produce an intermediate representation $H$. The kernel layer then transforms this correlation-encoded representation into a $d \times d$ weighted adjacency matrix. In fact, the output weighted adjacency matrix $A$ in equation (7) is literally the attention matrix in the last layer. Hence, the “key causal edge” in this context is equivalent to an edge in the graph with a large weight.
>
> Regarding the impact of head and layer numbers, in all examples we used only 1 head. We also found that the number of linear transformer layers directly affects the expressiveness of the ADAG model. When the number of training domains or variables increases, a more expressive model is required; therefore, in our experiments, we increase the number of layers from 10 to 40.

---

### Official Review · Reviewer_UPnk · 2025-10-30

**Soundness:** 1
**Presentation:** 2
**Contribution:** 2
**Rating:** 2
**Confidence:** 5

**Summary:**

This article builds a very limited foundation model, ADAG, for causal discovery. By assuming different causal structures of the same set of variables in different domains, ADAG conducts pre-training on a few domains to learn the shared features of causal structures among different domains, and then generalizes to other domains that have not been seen before. The test results show that ADAG can efficiently and accurately complete downstream multi-DAG learning tasks.

**Strengths:**

1. This paper utilizing a foundation model to directly mapping observed data to the causal graph, which can rapidly discovery causality and meet the requirements of real-world engineering applications.
2. This paper achieves small-sample causal discovery by training a foundation model, which has certain innovation.
3. The description in the method section of this article is clear.
4. When there are significant commonalities in across-domain data, the proposed model has been demonstrated to be effective and stable.

**Weaknesses:**

1. This paper presents a highly limited fundamental model for causal discovery, which can only learn linear causal relationships within the same set of variables, and may be far from real-world problems.
2. ADAG trains the foundation model by minimizing the reconstruction loss. Referring to previous causal discovery methods, although this strategy is theoretically effective, there is a risk of learning incorrect causal structures, which may seriously undermine the training of the foundation model.
3. Nonlinear expansion is immature. Simply concatenating nonlinear transformations with weighted adjacency matrices may cause the foundation model to fail to learn the mapping from observed data to causal structures.
4. The experimental results are confusing. In Figure 2, comparing the covariance matrix and principal components of the estimated A and the true A may not be as intuitive as directly comparing the two matrices. The caption and the figure of Figure 2 (b) seem inconsistent, which may mislead readers.
5. The experimental analysis is insufficient. In the experimental setup of this paper, general data is introduced, in which different domains have random DAGs. However, the experimental analysis overlooked how ADAG learns causal structure from general data. Given that the DAGs in different domains are random, ADAG may not be able to learn the shared features of causal structure from general data.
6. The efficiency and scalability of ADAG are limited. ADAG only offers slight improvements in accuracy and test efficiency, but it may have expensive training costs (under the same experimental conditions, it can only identify causal graphs with less than 50 nodes).
7. Real experiments might be unreasonable. This article sets that the variable set may have different causal structures in different domains. Take the dataset Sachs as an example. It contains multiple sub-datasets with a total of 7,746 observations. Different sub-datasets were observed after different interventions. Therefore, these sub-datasets can be regarded as coming from different domains and are highly suitable for verifying the effectiveness of ADAG. However, this paper only used one of the sub-datasets and was unable to verify ADAG's ability to identify causal graphs across domains. In addition, it is necessary to introduce other real datasets such as fMARI.
8. Lack of convincing theoretical proof. It seems that this paper merely references the identifiability analysis of existing methods, but lacks a mature theory to prove that ADAG can identify causal graphs and their weighted adjacency matrices that vary with the domain.

**Questions:**

1. Why refer to the mapping from the observed data to the weighted adjacency matrix as the kernel mapping? In my opinion, the foundation model of this article is merely a stack of Transformers and does not involve kernel operations.
2. Why say that strategies like DAGMA are not applicable to ADAG? Take DAGMA as an example. Although DAGMA uses the central path method for optimization, the more efficient acyclic constraints he proposed can also be optimized within the augmented Lagrange framework. I think methods like DAGMA do not conflict with the optimization process of ADAG.
3. Can you describe in more detail how ADAG extends to nonlinear structural equation models?
4. How does ADAG identify causal structures under general data? According to the description in this article, ADAG can learn the shared features of causal graphs in cross-domain data and then generalize to domains it has never seen before. However, the causal structure of general data in different domains is random. How does ADAG learn shared features from the random causal structure?
5. How long is the training time for ADAG? Although ADAG has improved testing efficiency, its training phase may consume a significant amount of resources and time. Since ADAG identifies linear causal relationships within the same set of variables, I think the expensive training cost might be unacceptable.
6. What are the metrics reported in Table 3? I think it's SHD, but this article doesn't explain the metrics reported in Table 3.
7. Can you validate the proposed model on other real-world datasets with domain-varying causal graphs?

---

> ### Author Response · Authors · 2025-12-02
> **Responses to reviewer UPnk (part 1)**
>
> We appreciate the reviewers’ time and effort, however it is clear that **fundamental aspects of our submission were misunderstood**, particularly regarding the architectural capabilities of our foundation model and the precise nature of our theoretical contributions. **These misunderstandings unfortunately led to an incorrect assessment of our empirical results.** In addition, some of the requests made in the reviews extend well beyond the scope of this paper and the boundaries of a typical conference submission. Hence, we respectfully request the reviewer to kindly reconsider their judgement given responses below, which provides r detailed clarifications and substantiates our claims.
>
> **Regarding Weakness 1.** The reviewer's comments appear to stem from a misconception regarding the scope and generalizability of our causal graph learning foundation model.  First, the assertion that our model is restricted to a fixed set of variables is incorrect and fundamentally misunderstands the concept of a foundation model in this context.  The very purpose of our proposed attention-based architecture is to learn robust, transferable patterns of variable dependency from observational data. Once trained on $d$ variables, the model is designed to generalize  to any system with fewer than or equal to $d$ variables,  only inferring the corresponding causal relations in the form of a DAG. Second, the critique of using **Linear SEMs** overlooks their **widespread acceptance and established efficacy** in the causal discovery community. The linear SEM setting is not only well-justified and prevalent in the causal discovery literature [1,2,3], but it also remains an active area of research. In particular, Dong et al. (2023) [4] has proven the success of linear SEMs in solving real-world problems across various domains. For instance, EEG/MEG source separation is routinely modeled using linear mapping functions. While we acknowledge that extending our framework to nonlinear settings represents a valuable avenue for future work, we must emphasize that our **core contribution** lies in demonstrating the **feasibility and efficacy** of a generalizable and unsupervised framework that can **infer DAGs**  on entirely new domains **without requiring any further training**.
>
> > [1] Zheng, Xun, et al. "Dags with no tears: Continuous optimization for structure learning." Advances in neural information processing systems 31 (2018).
>
> > [2] Park, Gunwoong. "Identifiability of additive noise models using conditional variances." Journal of Machine Learning Research 21.75 (2020): 1-34.
>
> > [3] Dai, Haoyue, Peter Spirtes, and Kun Zhang. "Independence testing-based approach to causal discovery under measurement error and linear non-gaussian models." Advances in Neural Information Processing Systems 35 (2022): 27524-27536.
>
> > [4] Dong, Xinshuai, et al. "A versatile causal discovery framework to allow causally-related hidden variables." arXiv preprint arXiv:2312.11001 (2023).

---

> ### Author Response · Authors · 2025-12-02
> **Responses to reviewer UPnk (part 2)**
>
> **Regarding Weakness 2.** We assume 'minimizing reconstruction loss' refers to our objective function: the mean squared error between observed data $X$ and the SEM reconstruction $XA$. The reviewer likely suggests a supervised approach with a different objective. However, employing data reconstruction error as the objective function offers a distinct advantage: it fully leverages the capacity of the linear SEM to model the data generation process itself. This approach is **unsupervised**: it operates without requiring ground-truth graph structures or prior knowledge of the causal mechanisms. This unsupervised setting of **causal discovery** is the default and often preferred setting in real-world applications, and it offers 4 advantages:
> - 1) **Data Availability & Scalability**: This is the **most realistic setting** because **ground-truth causal graphs** are virtually **never available** in real-world domains (e.g., biology, economics, climate science), a fact acknowledged by reviewers (Reviewer 5QJS and Reviewer qbVU). This fundamental design choice **significantly relaxes the data requirements** for foundation models, enabling learning across diverse domains where true causal graphs are unknown.
> - 2) **Generality & Flexibility**: Unsupervised methods are designed to infer causality from purely **observational data**. They do not require the costly, often impossible step of **human or experimental labeling**, making them broadly applicable across domains. This also results in lower sample requirements in testing as well, increasing its practical usages.
> - 3) **Theoretical Foundation**: as the reviewer noted, the entire field is built on the premise that causal information is embedded in the statistical properties of the joint distribution. Methods like Constraint-Based (e.g., PC, FCI) and Score-Based (e.g., GES) have rigorous theoretical guarantees (e.g., faithfulness, causal sufficiency) to recover the true Causal Graph or its Markov Equivalence Class under certain assumptions.
> - 4) **Focus on Data Generation process**: the objective function (e.g., minimizing reconstruction error, maximizing likelihood) directly models the data generation process (e.g., $X =XA + E$), aligning the learning goal with the scientific goal of understanding how the system works.
> In addition to the above 4 advantages, our proposed unsupervised approach also offers an additional benefit that is previously exclusive to the supervised approach:
> - 5) **Efficiency and Inference Speed**: once trained, our model can **instantly predict** the DAG for a new, unseen domain in a single forward pass. This is dramatically faster than the iterative search or independence testing required by most unsupervised algorithms.
> - 6) **Learning Complex Mappings between data and DAG**: furthermore, we demonstrate the novel capability of learning a complex, nonlinear mapping $f(X) \rightarrow A$, a feat previously thought possible only via supervision on the causal structure ($A$). Our architecture achieves this by a kernel function that maps raw data ($X$) directly to the causal graph ($A$), **without ever requiring thousands of labeled domain-graph pairs** ($X^\eta, A^\eta$). Crucially, we show that **this inference-only unsupervised transfer learning surpasses all tested state-of-the-art approaches on the testing datasets.** Regarding comment on the potential for incorrect causal structure learning in single-task DAG settings, we have provided **detailed theoretical justifications** in **Appendix D.1**. In summary, under certain conditions, the foundation model can robustly learn the nonlinear mapping between data $X$ and causal graph $A$.

---

> ### Author Response · Authors · 2025-12-02
> **Responses to reviewer UPnk (part 3)**
>
> **Regarding Weakness 3 and Question 3.** We must reiterate  that our focus on building foundation models for linear SEMs is **deliberate and justified**, as clearly stated in our paper and in response to Weakness 1. This choice is strongly supported by its prevalence in causal discovery literature and its established success in numerous real-world applications.
> Our inclusion of a discussion on a potential nonlinear extension and the **generalization ablation study** was intended precisely to demonstrate two key points: 1) Our underlying **framework is fundamentally extensible** to nonlinear settings. 2) The current linear foundation model exhibits a **reasonable degree of robustness** even when applied to nonlinear data.
> Regarding potential nonlinear formulations, multiple valid approaches exist. The formulation that combines nonlinear transformations with weighted adjacency matrices follows the method, as used by Yu et al. (2019) [5] (Eq. 3), , is  a widely adopted baseline for continuous causal discovery. **However, developing and rigorously testing a full nonlinear formulation of ADAG, complete with a novel associated learning algorithm, falls outside the scope of this particular work. We view Nonlinear ADAG as a highly promising and critical direction for future extension, and we welcome the opportunity to pursue this in depth in subsequent research. We review nonlinear ADAG as a promising direction for future extension, and we welcome the opportunity to pursue this in depth in subsequent research.**
>
> > [5] Yu, Yue, et al. "DAG-GNN: DAG structure learning with graph neural networks." International conference on machine learning. PMLR, 2019.
>
> **Regarding Weakness 4.** We note that the reviewer’s interpretation of Figure 2 is incorrect, probably due to the fact they did not understand the caption of Figure 2(b) and Figure 3. As we have pointed out in the caption of Figure 2(b): we are demonstrating the principal components of $A$ across multiple domains, NOT showing the learnt $A$ from one single domain. In the case of heterogeneous data generated with ER1 d=5 graphs, the space of $A$ from all domains is equivalent to $\mathbb{R}^5$, so the covariance of $A$ (treated as size-25 vectors) should have 5 non-zero eigenvalues. To evaluate the capability of ADAG in discovering this space of $A$, we apply the trained foundation model to 1000 test domains, obtain 1000 estimated adjacency matrices, compute their covariance matrix, and analyze its principal components. In Figure 2(a) we demonstrate the five non-zero eigenvalues from our estimated weighted adjacency matrices and the eigenvalues calculated from the ground truth $A$, and in Figure 2(b) the corresponding eigenvector to each eigenvalue is plotted. These results show that ADAG learns the correct shared features of causal structure: for any new and unseen domain, ADAG will predict the weighted adjacency matrices as a combination of these eigenvectors, that means, it discovers the correct low-dimensional space from multiple tasks, and explores in this space in new and unseen test domains. We also point out that a direct comparison of predicted $A$ and true $A$, as suggested by the reviewer, was already provided in Figure 3, along with the small reconstruction error, demonstrating that our method recovers the parameters of $A$ with high precision.

---

> ### Author Response · Authors · 2025-12-02
> **Responses to reviewer UPnk (part 4)**
>
> **Regarding Weakness 5 and Question 4.** As discussed above, the reviewer got this question possibly due to the misunderstanding on Figure 2. Firstly, we highlight that ADAG discovers the shared feature of $A$, not the shared feature of random data $X$. To understand the difference, one can start from the learning of estimator A in a basic linear regression example:
> $$(y^\eta(1:n),X^\eta(1:n)) \rightarrow A^\eta, \text{ where } A,X^\eta(i)\in \mathbb{R}^{d}, y^\eta(i)\in \mathbb{R}$$
> from data generated via:
> $$A^\eta \cdot X^\eta(i)=y^\eta(i)+\epsilon, A^\eta \sim \mathcal{N}(0,\Sigma_w),X^\eta(i) \sim \mathcal{N}(0,I_d),\epsilon\sim \mathcal{N}(0,\sigma^2).$$
> When $\Sigma_w$ has $w<d$ numbers of nonzero eigenvalues (for instance, when it is a diagonal matrix with the first $w$ diagonal elements $=1$ and the rest $=0$), $A^\eta$ has a low-dimensional structure, but $X$ does not. Even if the $A^\eta$ in the test domain is random, it still has a low-dimensional structure. This is the key for transformers to make zero-shot learning of estimator $A$ from a small number of $X(i)$ in the test domain.
> Second, the **ADAG model learns the low dimensional structure by constructing a complex, nonlinear mapping between the data $X$ and the causal graph $A$**. By training from multiple training domains, the underlying low-dimensional structure of $A$ is captured implicitly and applied as an empirical Bayes for the test domains. Still taking the linear regression example for simplicity, without the empirical Bayes, the learning problem of $A$ is ill-posed if $n<d$. However, with the correct prior of $A$ as $\mathcal{N}(0,\Sigma_w)$ inferred from training domains, the learning problem is well-posed if $w<n<d$.  In our work, with a sufficiently diverse training set spanning many distinct causal graphs, this mechanism enables the model to robustly infer a **unique causal graph** ($A$) from any new observation ($X$). We have clearly described the generation of this general data and demonstrated that our foundation model achieves **strong DAG learning accuracy** compared to state-of-the-art causal discovery methods.
>
> **Regarding Weakness 7 and Question 7.** Since we are the first to propose training a foundation model for DAG learning with zero-shot inference, there is no established protocol for evaluating such models on real-world data. Our motivation is to pre-train a general ADAG model that can infer any causal graph within a specified variable dimension, adhering to a "train-one-for-all" paradigm.. Accordingly, we adopt an evaluation strategy in which we train ADAG on synthetic general data and directly test it on observational samples from real-world datasets.
> For the Sachs dataset, although 7,746 samples are available, only 853 of them are purely observational. These 853 samples best align with the consensus ground-truth causal graph reported in the original paper, and evaluating methods on this subset has become a standard practice in many causal discovery works. We therefore follow this widely used protocol.
> We acknowledge that treating the sub-datasets as different “domains” and using them to train an ADAG model is an interesting alternative. However, this approach requires additional task-specific knowledge, particularly regarding how to define and select domains, and is not well aligned with our “train-one-for-all” paradigm. Under this paradigm, we focus on real-world datasets that provide both observational data and widely accepted ground-truth causal graphs. The Sachs and Sergio datasets best satisfy these criteria, and we believe our current evaluations on them sufficiently support our claims. fMRI suggested by the reviewer is also a time series dataset, beyond our scope of work.

---

> ### Author Response · Authors · 2025-12-02
> **Responses to reviewer UPnk (part 5)**
>
> **Regarding Weakness 6 and Question 5.** The characterization of our ADAG contributions as offering only "slight improvements" in accuracy and test-time efficiency is factually inaccurate and contrary to the evidence presented. As Figure 4 unequivocally demonstrates, ADAG achieves the best SHD across all evaluated settings, consistently outperforming state-of-the-art baselines. This consistent outperformance is evidence that ADAG effectively leverages the prior knowledge embedded in the cross-domain training data. In multiple key regimes, the improvements are substantial and highly consequential:
> - In challenging settings, such as dense graphs (ER4) and larger variable counts ($d=50$), ADAG achieves a markedly lower SHD than all competing methods.
> - The robustness under extremely limited data ($n=\{25, 50\}$) is particularly noteworthy. For the ER1, $d=5$ case, ADAG achieves an SHD of $\mathbf{0.0550}$. This represents a nearly $50\%$ improvement over the next-best competitor, MultiDAG ($0.1110$), and dramatically surpasses methods like NOTEARS ($0.6590$) and DAGMA ($0.6080$).
>
> Furthermore, Table 1 confirms that our test-time efficiency gains are not minor; they are orders of magnitude greater. ADAG's inference speed surpasses existing amortized methods such as AVICI and FIP by three orders of magnitude. Given these results—consistent SOTA accuracy across all regimes, substantial improvements in challenging settings, and massive gains in test-time efficiency—we maintain that the conclusion of only "slight" improvements is unsupported by the quantitative evidence.
>
> We understand the reviewer's concern regarding training cost, but we must emphasize that the **amortized-learning paradigm** employed by ADAG is the precise mechanism through which modern foundation models, including large language models, achieve both **scalability and low inference cost**. The necessity for upfront training resources is a well-accepted trade-off. Once trained, the foundation model exhibits **strong generalization** and can be applied to **any new dataset without costly retraining**. This is the core efficiency gain.
> Furthermore, a well-trained model for **small-to-moderate variable sizes ($d \le 50$)** is already powerful enough to enable **highly efficient inference of local structures** on variable subsets. These local inferences can then be composed into global graphs for much larger systems. Determining the existence of a viable kernel mapping between observations and weighted adjacency matrices is the **critical first step** for scaling this approach, which is why our initial experiments focus on this variable range.
> For the reviewer's reference, the specific training times for the general data setting ranged from approximately **13 minutes (for $d=5$) up to 40 hours (for $d=50$)**. Given the dramatic **three-orders-of-magnitude reduction in test-time inference cost** shown in Table 1, this upfront investment is demonstrably worthwhile. Finally, we must correct the reviewer's underlying assumption (related to Q5) about the ADAG model's scope:
> - **Variable Generalization**: Our ADAG model is **capable of far more than inferring DAGs for a fixed variable set**. As detailed in our response to Weakness 1, it is designed to generalize to any system with an equal or smaller number of variables than it was trained on.
> - **Nonlinear Robustness**: The model also demonstrates **reasonable generalization to nonlinear data**, as quantified in **Table 3**. This validates the foundational architecture's ability to extract causal information even when its core assumption (linear SEM) is violated, providing a strong basis for future nonlinear extensions.

---

> ### Author Response · Authors · 2025-12-02
> **Responses to reviewer UPnk (part 7)**
>
> **Regarding Weakness 8.** We provide a comprehensive theoretical justification that offers **clear intuition** for how ADAG learns the nonlinear mapping between data observations ($X$) and weighted adjacency matrices ($A$). Our framework is decomposed into two parts, and our theory addresses both:
> - **Single-Task DAG Learning**: We first state the necessary conditions on data distributions that **guarantee the identifiability of the linear SEM** for each individual domain task.
> - **Nonlinear Mapping Learning**: We then demonstrate that, provided the per-domain learned $A^\eta$ is correct for a **sufficiently large number of domains**, the nonlinear model can **accurately learn the mapping** from data to causal graphs, while being robust to a small number of erroneous domain estimates.
>
> While each component builds upon prior theoretical results, the **assembly of these components into a unified justification** for a **cross-domain, amortized causal discovery framework** like ADAG is **far from trivial** and constitutes a substantial theoretical effort.
>
> Furthermore, the reviewer overlooks our explicit and original theoretical contribution in Appendix D.2, where we go beyond graph identifiability and analyze parameter identifiability for the nonlinear mapping (between data and graph) learned by ADAG.
> - Parameter Identifiability: This analysis ensures that the inferred causal graph parameters are unique given the observational data.
> - Original Proof: To the best of our knowledge, ours is one of the very few works addressing this specific, high-level question, and the proof provided in Appendix D.2 is entirely our own contribution.
>
> Therefore, we fundamentally disagree with the claim that we merely cite existing results and lack original theoretical contributions; the rigorous analysis in Appendix D.2 provides novel, first-of-its-kind theoretical support for an amortized causal discovery model.

---

> ### Author Response · Authors · 2025-12-02
> **Responses to reviewer UPnk (part 8)**
>
> **Regarding Question 1.** The term "kernel mapping" does not refer to the explicit use of kernel operations (such as those found in Kernel PCA or Support Vector Machines) within the Transformer architecture itself. Instead, the term is used in its broader, more formal mathematical sense:
> - **The Adjacency Matrix as a Kernel**: In mathematics, a **kernel** is used to denote a weighting function for a weighted sum or integral (e.g., in integral equations or kernel methods). In the context of the Linear Structural Equation Model (SEM), the weighted adjacency matrix $A$ acts as the **set of coefficients or weights** that linearly combine the variables' error terms ($E$) and the observed data ($X$) to define the causal relations: $X = XA +E$. Therefore, $A$ functions as the **kernel of the vector space $\mathbb{R}^d$** that governs the data generation process.
> - **The Foundation Model's Objective**: Our foundation model is designed to learn the complex, **nonlinear mapping** $f: X \rightarrow A$.
> - **Conclusion**: Hence, our method aims to provide a nonlinear mapping from the **observed data ($X$) to the underlying kernel (the weighted adjacency matrix $A$)** that defines the system's causal structure.
>
> This usage emphasizes that the model is predicting the core **weighting mechanism** (the kernel) of the causal system, rather than just an arbitrary matrix.
>
> **Regarding Q6.** The metrics reported in Table 3 is the SHD.

---

### Author Response · Authors · 2025-12-02
**Summary of the reviews**

We are grateful to the reviewers for their constructive feedback and for recognizing the key strengths of our work, including:
- (i) the novelty of our attention mechanism-based model to learn DAG in an unsupervised manner, and its immense practical value (Reviewer UPnk, Reviewer 5QJS, Reviewer qbVU, Reviewer XKxT);
- (ii) substantial improvement against all eight SotA baselines, alongside with theoretical analysis on parameter identification （Reviewer 5QJS, Reviewer qbVU, Reviewer XKxT);
- (iii) the comprehensive and multifaceted evaluation across both synthetic and real-world datasets (Reviewer UPnk, Reviewer 5QJS).

Key issues clarified during rebuttal include:
- Further clarification on the advantages of unsupervised learning DAG foundation model: we discussed **four advantages** of unsupervised approach over supervised DAG learning approach (namely, data availability, flexibility, theoretical foundation, and generative process),  and also highlighted two **major contributions** of our method to the unsupervised learning approach (namely, efficiency and learning the mapping from data to graphs), which has not be done before. ( Reviewer UPnk and Reviewer XKxT)
- Justification on linear SEM choices, by noting it is an active research area and many potential applications are based on the linear formulation. (Reviewer UPnk, Reviewer qbVU, and Reviewer XKxT)
- We provided **further theoretical and practical clarification** on how our method works, addressing the common misconceptions raised by all the reviewers. (All reviewers)

We are confident that we have **fully addressed all concerns** raised by the reviewers, many of which stemmed from initial misunderstandings about the proposed approach. We thank the Area Chairs for their time, consideration, and for granting the additional time for this response.

---

### Meta-Review · Area_Chair_tCQh · 2025-12-31

**Summary:**

Causal learning has recently been something of a hot topic. The commonly used methods for learning the underlying DAG do not scale beyond 50-100 vertices, unfortunately. The authors try to address this scalability issue by pre-training a "foundational model" for learning DAGs. This is novel and most of the reviewers（Reviewer 5QJS, Reviewer qbVU, Reviewer XKxT) agree that this seems to perform rather well, in terms of statistical performance, but does not scale beyond the 50-100 vertices, yet. Unfortunately, the reviewers were not enthusiastic.

**Reviewer Concerns:**

Reviewers' key concerns included:

- questions of identifiability. While in some cases, one may identify only a Markov Equivalence Class, rather than a unique DAG, and this has not been well addressed in the original submission, this does not make the computational performance any less impressive.

- linear SEM vs. non-linear SEM. As has been clearly argued in the rebuttal, linear SEM is of substantial interest.

- theoretical justification. Considering the nature of the approach utilized, the AC finds it surprising any analysis is possible at all, as has also been clearly argued in the rebuttal.

The AC is concerned about the reproducibility of the results, considering there is no supplementary material, no huggingface/anonymous github linked etc. Also, the AC is concerned that none of the reviewers inquired into the reproducibility.

**Reviewer Scores:**

I believe that reviewers' key concerns have been well addressed. I would imagine that the reviewers could well improve their scores somewhat.

---

### Decision · Program_Chairs · 2026-01-26

Reject